# Delivered complementation in planta (DCIP) enables measurement of peptide-mediated protein delivery efficiency in plants

Jeffrey W. Wang [1], Henry J. Squire [1], Natalie S. Goh[1], Heyuan Michael Ni [2], Edward Lien[2], Cerise Wong[1], Eduardo González-Grandío [3] & Markita P. Landry [1,4,5,6 ✉]

Using a fluorescence complementation assay, Delivered Complementation in Planta (DCIP), we demonstrate cell-penetrating peptide-mediated cytosolic delivery of peptides and recombinant proteins in Nicotiana benthamiana. We show that DCIP enables quantitative measurement of protein delivery efficiency and enables functional screening of cell-penetrating peptides for in-planta protein delivery. Finally, we demonstrate that DCIP detects cell-penetrating peptide-mediated delivery of recombinantly expressed proteins such as mCherry and Lifeact into intact leaves. We also demonstrate delivery of a recombinant plant transcription factor, WUSCHEL (AtWUS), into N. benthamiana. RT-qPCR analysis of AtWUS delivery in Arabidopsis seedlings also suggests delivered WUS can recapitulate transcriptional changes induced by overexpression of AtWUS. Taken together, our findings demonstrate that DCIP offers a new and powerful tool for interrogating cytosolic delivery of proteins in plants and highlights future avenues for engineering plant physiology.

[1] Department of Chemical and Biomolecular Engineering, University of California, Berkeley, Berkeley, CA 94720, USA. [2] Department of Molecular and Cell Biology, University of California, Berkeley, Berkeley, CA 94720, USA. [3] Plant Molecular Genetics Department, Centro Nacional de Biotecnología-CSIC, Campus Universidad Autónoma de Madrid, Madrid, Spain. [4] Innovative Genomics Institute, Berkeley, CA 94720, USA. [5] California Institute for Quantitative Biosciences, University of California, Berkeley, Berkeley, CA 94720, USA. [6] Chan Zuckerberg Biohub, San Francisco, CA 94063, USA. ✉email: landry@berkeley.edu

The delivery of proteins to walled plant cells remains an ongoing challenge. In addition to the cell membrane, the plant cell wall is an effective cellular barrier not only to naturally occurring pathogens but also to introduce macro-bio-molecules: DNA, RNA, and proteins. While several tools exist for the delivery of nucleic acids in plants, very few enable the delivery of proteins to walled plant cells. The development of CRISPR-Cas9[1] and other DNA editing tools[2] has only increased the need for working protein delivery tools, which could accelerate basic research, spawn novel agricultural biologic agents, or potentiate DNA-free gene editing of plants. Recent discoveries in morphogenic transcription factors that accelerate plant regeneration evince a new class of possible protein cargoes if these proteins could be delivered[3]. These motivations have led researchers to develop novel nanoparticle-based strategies for the delivery of biomacromolecules to walled plant cells. For example, multiple technologies have been developed to deliver siRNA to plants using diverse vehicles such as single-walled carbon nanotubes[4], DNA nanostructures[5], carbon dots[6], and gold nanoparticles[7]. Although the cell wall may be permissible to materials below the size exclusion limit of 5–10 nm or proteins around 50–100 kDa[8], fewer have demonstrated delivery of proteins using cell-penetrating peptides[9,10]. Despite these proof-of-principle advances, protein delivery to walled plant cells remains largely dependent on biolistic delivery, which requires protein dehydration (and thus potential inactivation) to a gold particle surface and forceful and injurious rupture of plant membranes to accomplish delivery in a low throughput and low-efficiency manner[11]. One main barrier to the use of nanotechnologies for plant biomolecule delivery, and specifically cell-penetrating peptides for protein delivery to plants, is the lack of quantitative validation of successful intracellular protein delivery. This barrier makes it difficult to unilaterally distinguish successful protein delivery from artifact, lytic sequestration, or quantitatively compare the delivery efficiency of different peptides[12].

This lack of tools to quantify successful protein delivery in plants is due to the near-universal dependence on confocal microscopy to validate the delivery of fluorescent proxy cargoes. However, confocal microscopy in plant tissues poses a set of unique problems that make it challenging to distinguish artifacts from signals and make absolute quantification of signals impossible. Aerial tissues of plants are heterogeneous, highly light scattering, and possess intrinsic auto-fluorescence[13], which makes it difficult to distinguish signal from noise. Furthermore, unlike mammalian cells, the plant cell cytosol in the majority of cell types is highly compressed against the cell wall by the plant's large central vacuole, making unambiguous imaging of cytosolic contents challenging due to the small surface area of cytosolic contents[14]. In addition, the plant cell is surrounded by a porous and adsorbent cellulosic wall that is 100–500 nm thick[15] which spans the Rayleigh diffraction resolution limit of visible light imaging and the axial resolution of most confocal microscopes. Together, the small cytosolic volume which is proximal to the cell wall makes it impossible to distinguish—with the necessary spatial precision—the location of fluorescent cargoes near versus imbedded in the cell wall, or inside the cell cytosol[7], without super-resolution microscopy[16]. Additionally, free fluorophore from cargo degradation[17] or endosomal entrapment of cargoes would contribute to measured fluorescence intensity and intracellular colocalization in plants but fail to correlate with successful delivery. For these reasons, gauging cellular uptake of cargoes based solely on confocal microscopy data of fluorophore-tagged cargo in plants does not confirm successful intracellular delivery nor provide quantitative data for effective uptake. These barriers have made biomacromolecule delivery in plants, particularly protein delivery, exceptionally challenging. We, therefore,

developed a versatile, unambiguous platform to confirm the delivery of proteins in walled plant tissues and demonstrate it is possible to quantify protein delivery efficiency across different protein sizes.

We designed an *Agrobacterium tumefaciens* expression mediated, GFP-complementation based, red/green ratiometric sensor for the detection of protein delivery in plants using confocal microscopy (DCIP, *D*elivered *C*omplementation *in* pl*a*nta). In this technique, sfGFP is split between a larger non-fluorescent fragment (sfGFP1–10) and a smaller peptide strand (GFP11)[18,19]. When GFP11 is delivered to the same compartment as sfGFP1–10, only then is GFP fluorescence reconstituted (Fig. 1a). This method has the critical benefit of only producing a signal if the peptide tag remains intact, is successfully delivered to the cytosol, and is not sequestered in lytic organelles or trapped in the apoplast. GFP11 also serves as an excellent reporter tag because its short length (16AA) is accessible to chemical synthesis and because it is readily incorporated into recombinant proteins as a terminal tag. The final design of DCIP was also guided by the desire to perform automated image analysis within complex leaf tissues and thus reduce the risk of bias during analysis.

While bimolecular fluorescence complementation and nanoluciferase complementation have been previously applied for measuring cell-penetrating peptide (CPP)-mediated delivery in mammalian cells[20–22], it has not been employed to confirm the delivery of bio-cargoes to plant cells. CPPs are small cationic or amphipathic peptides that when conjugated to cargoes, enable cytosolic delivery[23]. We utilized CPPs to test DCIP due to their synthetic accessibility, previous deployment in plant-tailored delivery schemes[24,25], and because much of their underlying cell-penetrating mechanisms in plants remain unstudied. In doing so, we show that DCIP can confirm the delivery of as little as 10 µM of GFP11 peptide via confocal imaging and that nearly comprehensive (>78%) delivery to leaf cells is achieved with 300 µM of CPP-delivered GFP11. Furthermore, we used DCIP to quantify the relative effectiveness of several popular CPPs (TAT, R9, BP100) to accomplish protein delivery in plants and reveal that R9-mediated delivery in leaves is largely independent of endocytosis. We also used DCIP to probe the stability of R9-GFP11 in leaf tissues and show a concomitant disappearance of tissue-localized GFP11 and GFP complementation signal within 24 h. Finally, we utilize DCIP to demonstrate the CPP-mediated delivery of several recombinant proteins into the cytosol of mesophyll and pavement cells. As a proof of concept, we also use DCIP to show that CPP-based delivery can be used to form novel protein–protein interactions in live plants and demonstrate the delivery of *Arabidopsis* WUSCHEL transcription factor, evincing the possibility of physiologic engineering of plants using delivered proteins or peptides.

## Results

**Design of the delivered complementation *in planta* (DCIP) sensor system and workflow**. Although GFP bimolecular fluorescence complementation has been used to detect CPP-mediated delivery in mammalian cells, it has not yet been employed in plants; where it is arguably most beneficial to confirm successful biomolecule delivery. We, therefore, developed an imaging-based strategy for quantifying delivery-mediated GFP complementation with DCIP. Advantages of an image-based approach include the ability to assess delivery in the complex structure of leaves and the removal of ambiguity caused by total lysis as required by a luciferase-complementation-based approach[22].

The delivery sensor protein (DCIP) consists of sfGFP1-10 C-terminally fused to mCherry with an N-terminal SV40 NLS[26] and is transiently expressed in leaves using agrobacterium

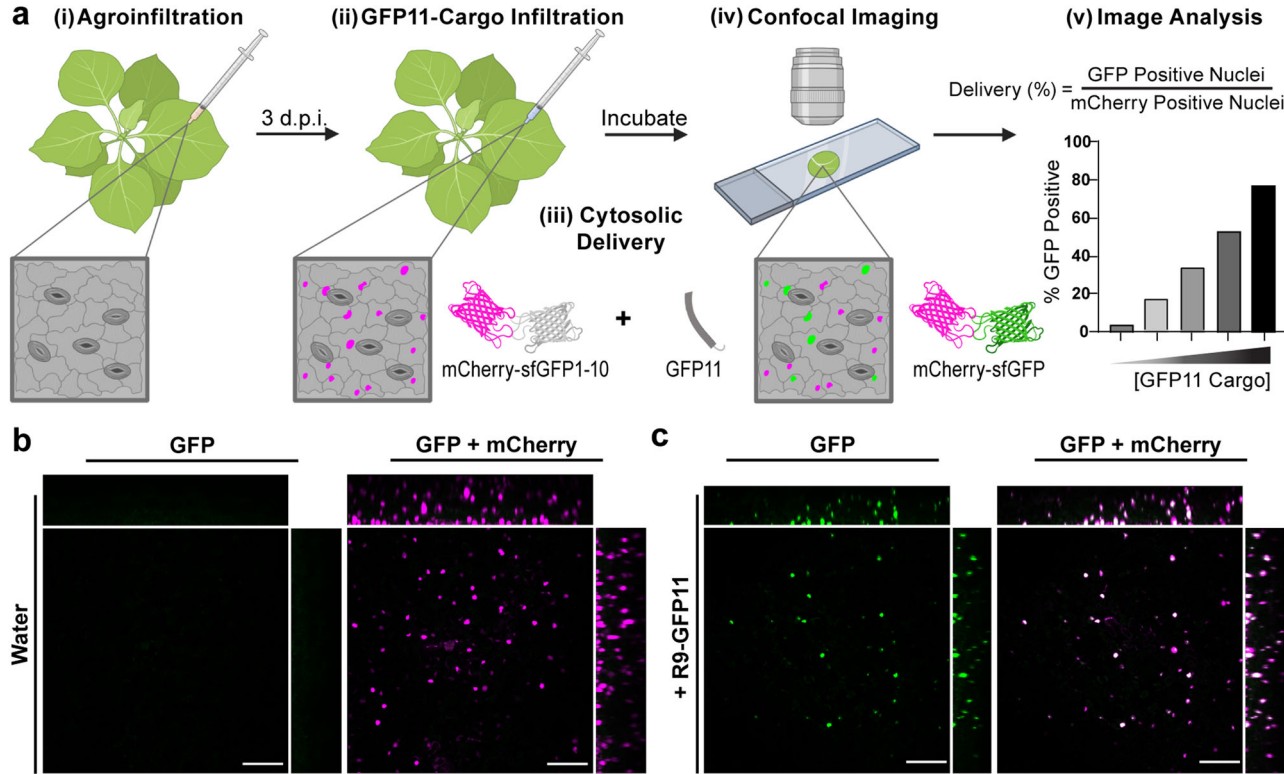

**Fig. 1 Demonstration of DCIP for peptide delivery. a** Workflow schematic (created with BioRender.com) for DCIP. (i) *N. benthamiana* is agroinfiltrated with the DCIP vector. (ii) 3 d.p.i. plants are infiltrated with the peptide or protein cargo fused to GFP11. (iii) During incubation, GFP11 is internalized into plant cells and if cytosolic delivery occurs, GFP11 is able to complement GFP1-10 and sfGFP fluorescence is recovered. (iv) Post-incubation, leaf discs are imaged and analyzed using Cell Profiler by using mCherry fluorescence to identify cells via their fluorescent nuclei. (v) The sfGFP fluorescence is normalized to mCherry fluorescence to account for variability in DCIP expression and the number of GFP-positive cells relative to the total number of mCherry positive cells is determined as an analog to delivery efficiency. **b** Representative maximum intensity projection of a leaf disc expressing DCIP infiltrated with water. sfGFP fluorescence is pseudocolored green (left) and two-color overlay with mCherry fluorescence, pseudocolored magenta, resulting in a white appearance after overlay (right). mCherry-expressing cells possess nuclei presenting as small, round fluorescent bodies amenable to automated image analysis. Orthogonal projections demonstrate the depth of imaging in leaves. **c** Equivalent images of DCIP-expressing leaf after treatment with 100 μM R9-GFP11 for 4 h and showing that delivery capability extends throughout the full thickness of the leaf tissue. Scale bar represents 100 μm.

(Supplemental Fig. S1). For DCIP, we chose a mCherry fusion for three reasons: (1) mCherry is easy to spectrally resolve from plant autofluorescence (2) a constitutive fusion allows identification of positively *A. tumefasciens* transfected cells and (3) mCherry fusion permits ratiometric quantification of GFP bimolecular fluorescence complementation since the relative expression of sfGFP1-10 is tied to the expression of mCherry by direct fusion. Because plant cells are heterogeneous in shape and have many autofluorescent bodies, we localized the sensor to the nucleus to produce a round, uniform object that is amenable to automated image analysis and provides unambiguous confirmation of successful delivery of GFP11 or GFP11-tagged cargoes. We also hypothesized that the NLS localization of DCIP should allow the detection of a broad range of sizes of delivered cargoes as the size exclusion limit for efficient transport through the nuclear envelope is presumed to be >60 kDa[27]. We constructed the coding sequence of the DCIP sequence by traditional restriction ligation cloning and the final transcriptional unit assembly was performed using Goldenbraid 2.0[28]. In tandem, we also developed a cytosolically localized version of DCIP, cytoDCIP, which lacks SV40 NLS. These constructs were transformed into *A. tumefaciens* and agroinfiltrated in *Nicotiana benthamiana* plants. *N. benthamiana* was chosen as a model plant due to its common use in transient expression experiments as well as in delivery experiments[29]. Successful expression of intact DCIP three days post agroinfiltration (d.p.i.) was verified by microscopic

observation (Fig. 1b) and by Western blot using an anti-mCherry antibody (Supplemental Fig. S2), and prior to attempts at GFP11 or GFP11-tagged cargo delivery.

A typical experimental workflow using DCIP is provided in Fig. 1a. The DCIP protocol involves transient expression of DCIP in *N. benthamiana*. 3 d.p.i., leaves are infiltrated with an aqueous solution of cargo that contains the GFP11 tag. Immediately after infiltration, the infiltrated leaves are either left intact or a leaf disc is excised from the infiltrated area and plated on pH 5.7 ½ MS. After a predetermined incubation time, the leaves are imaged on a confocal laser scanning microscope. For quantitative imaging, the resulting images are then automatically analyzed using Cell Profiler[30] for nuclear sfGFP and mCherry fluorescence (Fig. 1a). A detailed methodology is provided in the "Methods" section.

For the first proof of concept, we used the nona-arginine (R9) cell-penetrating peptide fused to GFP11 by a (GS)₂ linker to validate the functionality of DCIP to detect successful GFP11 delivery. R9 was chosen due to its known effectiveness in both plant[24] and mammalian[31] systems as well as its relatively well-characterized mechanism of action in mammalian cells[32]. Without infiltration or infiltration with water, no sfGFP fluorescence is observed (Fig. 1b) and only mCherry-containing nuclei can be seen. Upon infiltration with 100 μM R9-GFP11, we observed robust sfGFP complementation at 4–5 h post-infiltration that colocalized with mCherry (Fig. 1c). The timing of 4–5 h was determined by balancing the reported mammalian uptake kinetics

for R9 peptides (<1 h)[31,33] against the relatively slow process of GFP complementation which required >5 h for total complementation in our hands using an in vitro system with recombinant sfGFP1–10 (Supplemental Fig. S3a). For comparison, previous dye-labeled cargo CPP-mediated delivery experiments in plants often used time points of about 2 h[24]. In this case, an 8 mm leaf disc was excised from peptide-infiltrated tissue and plated onto ½ MS to control possible apoplastic flow and uncontrolled drying of the infiltrated liquid which may change the effective concentration of R9-GFP11 the cells experience. An orthogonal projection (Fig. 1c) shows GFP complementation deep (~100 μm) into the z-axis of the leaf in both pavement cells and mesophyll cells. Imaging at a lower magnification shows efficient delivery throughout the leaf disc using DCIP (Supplemental Fig. S4).

**Validating DCIP to quantify protein delivery efficiency using R9-GFP11**. We next assessed whether DCIP would be able to quantify the relative effectiveness of peptide delivery in planta. We once again used R9-GFP11 for validation experiments and infiltrated a range of concentrations from 0 to 100 μM R9-GFP11 into DCIP expressing N. benthamiana. The initial concentration range was determined from previously reported effective concentrations for mammalian cells[31,34]. After 4–5 h of incubation, we observe a clear concentration-dependent upshift of the green/red ratio in DCIP expressing nuclei (Fig. 2a). Seven biological repeats (one plant per repeat) were acquired using this methodology. The resulting mean green/red ratio from each repeat was averaged and show a statistically significant ($p < 0.05$) increase of green/red at concentrations greater than or equal to 50 μM R9-GFP11 (Fig. 2b). Further gating for percent GFP-positive nuclei to account for variations in quantitating fluorescence data in microscopy also show significant ($p < 0.05$) delivery at 20 μM (Fig. 2c). We also observe slightly improved linearity of response with respect to R9-GFP11 concentration using percent positive instead of averaged green/red ratio. We then defined delivery efficiency by normalization of the percentage of GFP positive cells with respect to the 100 μM treatment (Fig. 2d). Normalization helps reduce the biological variation in the absolute uptake efficiency we observed (Supplemental Fig. S5). We attribute this lack of lower-end sensitivity below 20 μM to the erroneous detection of autofluorescent bodies in the 0 μM control used for thresholding. From this initial performance study, we observed that relatively higher concentrations of R9-cargo are required for efficient delivery when compared to mammalian cells which can undergo delivery at low micromolar concentrations[21]. However, we also observed that relatively high delivery efficiency (>40% positive) could be achieved in plants when leaves were treated with 100 μM R9-GFP11 and greater (Fig. 2c).

After a low concentration-range validation, we sought to determine at what point the DCIP signal saturates, representing the maximal possible protein delivery efficiency in plant leaves. With the aforementioned workflow, DCIP-expressing leaves were infiltrated with 0–1000 μM R9-GFP11 in water and incubated as leaf discs for 4-5H. Once again, we observe strong upshifting of the green/red ratio of DCIP nuclei in treated leaves as a function of R9-GFP11 concentration (Fig. 2e). After six experimental repeats (one plant per repeat), we also observe that concentrations greater than 300 μM R9-GFP11 show a statistically significant increase in delivery compared to 100 μM R9-GFP11 and a relative saturation in green/red ratio at concentrations at 300 μM and higher (Fig. 2f). Similarly, the data analyzed using percent GFP positive cells show a saturation at about 75% positive at 500 μM and above (Fig. 2g), while the difference is significant at 300 μM if the delivery efficiency is normalized to the 100 μM

control to account for inter-plant variation in delivery (Fig. 2h). Examples of maximum intensity projections of the GFP channel for a titration DCIP experiment are provided in Fig. 2g. Importantly, although quantitation failed to detect statistical significance across samples treated with less than 20 μM R9-GFP11, successful delivery could still be occasionally observed for these low peptide concentrations as sfGFP complemented nuclei (Fig. 2i). Multi-channel images showing mCherry expression show strong colocalization of the sfGFP signal and mCherry nuclear signals at all tested concentrations (Supplemental Fig. S6).

**Assessing CPP performance and mechanism with DCIP**. After validating DCIP for quantitative peptide delivery, we assessed whether DCIP could be used to screen for effective cell-penetrating peptide sequences in leaves. Three commonly used cell-penetrating peptide sequences were assessed (Fig. 3a). BP100 is a microbially derived CPP that has been previously reported to be effective in plants through a dye conjugation and delivery experiment[24]. TAT is an arginine-rich HIV-1 derived peptide and one of the first cell-penetrating peptides characterized[35]. R9 is a derivative of TAT where all amino acids are substituted for arginine[31]. Each of these CPPs was produced through solid-phase synthesis as fusions to GFP11, separated by a short (GS)$_2$ linker. We used an in vitro bimolecular fluorescence complementation assay to ensure that the CPP fusions did not interfere with complementation activity (Supplemental Fig. S3a). After infiltrating 100 μM of each peptide construct into DCIP expressing leaves and incubating for 4-5H, confocal image analysis revealed that both TAT and R9 were effective at delivering GFP11 into plant cells and enabled delivery efficiencies ranging from 30–80% (Fig. 3b). R9 appeared to be the most effective of the tested CPPs with TAT being 0.87 times as effective as R9, and BP100 or GFP11 alone showing no statistically significant signal (Fig. 3c). These results support that without a CPP, GFP11 is not able to enter the cytosol of plant cells. To our surprise, BP100-mediated delivery was not statistically significantly better than either the water infiltration control or GFP11 alone. Once again, although BP100 was not statistically significantly better than the water control or GFP11 alone, we were able to observe rare instances of successful delivery for 100 μM BP100-GFP11 (Supplemental Fig. S7) but not for the negative control or GFP11 alone. Closer inspection of the imaged nuclei also revealed strong nucleolar localization of sfGFP in TAT-GFP11 and R9-GFP11 treatments (Fig. 3d). These images suggest that R9 and TAT remain intact when bound to sfGFP1-10 in the cell, as poly-arginine motifs are known to localize to the nucleolus[36].

After validation of R9-GFP11 as the best-performing CPP for protein delivery in plants, we sought to probe the mechanism by which R9 delivers cargo to the plant cell. Specifically, we probed whether R9 delivery was endocytosis dependent or independent. The delivery efficiency (based on percentage of positive cells) in leaf discs infiltrated with 100 μM R9GFP11 and incubated at 4 °C or room temperature was not statistically different ($p > 0.05$) (Fig. 3e) regardless of normalization with respect to the room temperature control (Fig. 3f). However, the normalized delivery intensity, as defined by green/red ratio normalized to the room temperature treatment showed that the 4 °C discs possessed lower sfGFP fluorescence (Fig. 3g). This aligns with our in vitro data showing that bimolecular fluorescence complementation is possible at 4 °C, although compromised in efficiency (Supplemental Fig. S3b). These data suggest that R9 delivery is largely independent of cellular activity such as endocytosis. Co-infiltration of R9-GFP11 and endocytosis inhibitors wortmannin or ikarugamycin[37–39] similarly resulted in no statistically significant decrease in delivery efficiency (Fig. 3h). Taken

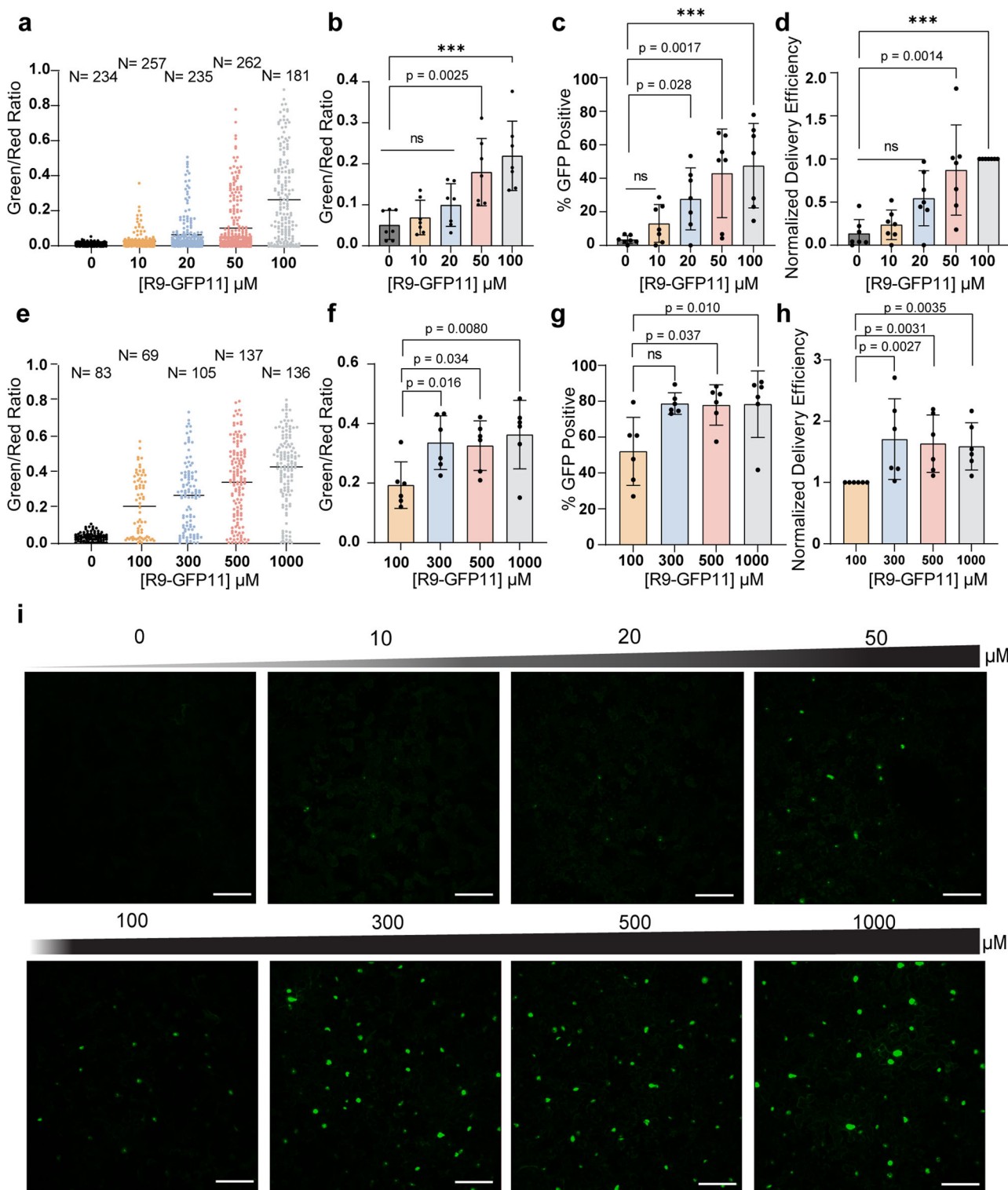

together, these data align with previously reported studies in mammalian cells and a singular plant protoplast centered study[40] that suggest at concentrations >10 µM, R9 enter cells through a combination of direct membrane permeation and through endocytosis and subsequent endosomal escape[31,32].

**The stability of peptide cargoes in leaves probed using DCIP.** We next confirmed that DCIP is capable of detecting successful delivery in attached, intact leaves instead of leaf discs. DCIP-expressing leaves were thus infiltrated with 100 µM R9-GFP11

solutions and allowed to incubate in situ. During this time, the infiltrated liquid would dry, rendering the effective treatment concentration difficult to compare to the above leaf disc assays. We then assessed the amount of sfGFP complementation at 4 or 24 h post infiltration with 100 µM R9-GFP11 (Fig. 4a). In accordance with the previous leaf disc assays, we observed a relatively high percentage of sfGFP positive cells at 4 h post infiltration in intact leaves. Conversely, at 24 h post infiltration, the percentage of sfGFP-positive cells returned nearly to baseline and the percent of GFP-positive cells was not statistically different

**Fig. 2 Validation of DCIP using concentration titrations of R9-GFP11. a** Representative green/red ratio in *N. benthamiana* expressing DCIP infiltrated with 0–100 μM R9-GFP11 or a water control for 4–5 h. Each point represents the relative fluorescence from sfGFP caused by delivered complementation and mCherry expression in a single nucleus. Successful delivery of the GFP11 cargo results in an increase in the green/red ratio. **b** Mean green/red ratio averaged across seven plants as experimental repeats ($N = 7$). Error bars represent the standard deviation of the group repeats. Statistical comparisons between each treatment condition and the non-treated control with a Kruskal–Wallis test in combination with Dunn's multiple comparison test. **c** Percentage of GFP-positive cells averaged across $N = 7$ biological repeats and, **d** corresponding delivery efficiency normalized to the 100 μM treatment group. **e** Representative green/red ratio after treating leaf discs with 100–1000 μM R9-GFP11 for 4–5 h. **f** Mean green/red ratio averaged across six biological repeats ($N = 6$) after treating leaf discs with 100–1000 μM R9-GFP11 for 4–5 h. **g** Average percent GFP-positive cells and, **h** calculated normalized delivery efficiency of leaves treated with 100–1000 μM R9-GFP11 for 4–5 h ($N = 6$). Exact *p*-values are given for $0.001 < p < 0.05$, ***$p < 0.001$ and, ns = $p > 0.05$. **i** Representative single color maximum intensity projections of leaves infiltrated with 0–1000 μM R9-GFP11 and incubated for 4–5 h. Nuclei exhibiting delivered complementation appear as round, green objects. Scale bar is 100 μm. All error bars are standard deviations.

than the percent positive in the non-treated control. We also tested whether or not 24 h-treated cells were still competent to R9-mediated delivery; we retreated a previously 24H treated leaf with 100 μM R9-GFP11, and imaged 4 h post re-treatment. In this re-treatment, we visually identified some instances of successful delivery (Fig. 4b). The limited fluorescence recovery with re-treatment could be due to an induced stress response that decreases the cell wall permeability[41]. These data show that the signal observed from GFP11 delivery is transient but can be somewhat recovered with re-treatment.

After DCIP revealed the transient delivery response, we sought to probe whether the tissue stability of the R9-GFP11 constructs correlates with the rapid peak and decline in DCIP response. We infiltrated leaves with 500 μM R9-GFP11 and left the plants to incubate in situ. A higher concentration of R9-GFP11 was chosen to facilitate detection by immunoblotting. After 0, 4, 8, and 24 h of incubation, a single 12 mm leaf disc was harvested and lysed for each treatment. Two microliters of lysates were then spotted onto nitrocellulose for dot blot analysis[42] using a primary antibody raised against GFP11. The dot blot analysis shows rapid instability of GFP11 peptides in leaf tissues, with the quantity of recovered R9-GFP11 returning to levels of the non-treated control by 24 h (Fig. 4c, d). These data suggest that the observed disappearance of the DCIP signal is connected to the clearance of the delivered peptide in the leaf tissue. The reduction at 24 h in the sfGFP signal is indicative of the combined intracellular and extracellular turnover of GFP11 becoming higher than the rate of delivery and complementation. These data are in line with the limited serum stability of R9-conjugates in mammalian systems in which low peptide stability was associated with extracellular proteases and intracellular clearance of the peptide-cargoes[43,44]. We hypothesize that similar dynamics from apoplastic proteases[45] and intracellular degradation are at play in plants. We would also add the additional caveat that the changes in DCIP data may also be partially due to the intracellular half-life (~26 h) of GFP[46]. Finally, to demonstrate the importance of the R9-GFP11 linkage in successful delivery, we infiltrated plants with free GFP11 and free R9 peptide at a 1:3 or 1:1 molar ratio. Delivery did not occur when R9 was not covalently conjugated to GFP11 (Fig. 4e).

**DCIP enables detection of successful delivery of recombinant proteins.** While the above quantification of CPP-delivered peptides is useful, the delivery of larger constructs is required to facilitate the major goals of plant delivery. This motivated our design of a ligation-independent cloning[47]-based *E. coli* expression vector for purifying recombinant proteins tagged with an N-terminal GFP11 and a C-terminal R9 peptide (Fig. 5a). To test the vector, we used mCherry (26.6 kDa) as a model protein. Although mCherry fluorescence overlaps with DCIP and precludes quantitative imaging, the identification of sfGFP fluorescent nuclei would confirm successful delivery. Furthermore, the

innate mCherry fluorescence offers a glimpse of how well-distributed the infiltrated protein solution is in the leaf tissue. We also purified an additional mCherry fusion cloned with a 3′ stop codon that would prevent tagging with the C-terminal R9 as a control. The total construct molecular weights were 32 kDa without R9 and 34.5 kDa with R9 and verified by SDS–PAGE (Supplemental Fig. S12a).

As can be seen in Fig. 5b and e, infiltration with 60 μM of either mCherry construct results in a thorough coverage of mCherry fluorescence. In GFP11-mCherry-R9 imaging, we were unable to discern most nuclear localized DCIP mCherry from the infiltrated mCherry due to the bright signal of the infiltrated protein. Despite this, some nuclei possessing both DCIP expression and GFP fluorescence can be observed with close examination (Fig. 5h). Clear DCIP expression can be seen in the GFP11-mCherry treatment but without GFP fluorescence (Fig. 5f). This infiltration experiment also demonstrates the subjectivity of determining internalization from tissue-dispersed cargo fluorescence as the mCherry channel shows little obvious difference in appearance with or without R9 (Fig. 5b and e). Only with a careful inspection can delivery be confirmed in the mCherry-R9 case by observing mCherry excluded zones caused by plastids (Fig. 5h, i).

Using the sfGFP channel, bright green fluorescent nuclei can be observed exclusively in the GFP11-mCherry-R9 infiltration (Fig. 5c). In addition to nuclei, we also observed numerous punctate sfGFP fluorescent objects that we hypothesize could be aggregates or partially entrapped GFP11-mCherry-R9 that have successfully undergone bimolecular fluorescence complementation. The observation of partial endosomal entrapment of mCherry aligns with the observation in mammalian cells that larger R9-delivered cargoes may undergo endocytosis and entrapment[48]. In contrast, without R9, no sfGFP fluorescence was observed (Fig. 5f), further confirming the successful delivery of mCherry only when tagged with R9. An overlay of all channels with a chloroplast channel shows thorough delivery throughout the leaf tissue and that the green fluorescence does not arise from plastid autofluorescence (Fig. 5d and g). We also observed that successful delivery of the larger mCherry cargo appears qualitatively less efficient than the delivery of smaller peptides (Supplemental Fig. S8).

We next attempted to deliver a GFP11-BFP-R9 construct (Supplemental Fig. S9a). However, we found that R9 fusion generated BFP that was practically insoluble at a physiologic pH of 5.7–7.5[49] and was only soluble at pH 10.8 (Supplemental Fig. S9b). Regardless, we infiltrated a DCIP-expressing plant with a mixture of insoluble protein at pH 9.0 at an initial protein concentration of 100 μM. Indeed, we see rare instances of GFP complementation using GFP11-BFP-R9 (Fig. 5j). A profile intensity analysis of GFP fluorescent nuclei (Fig. 5k and l) shows colocalization of mCherry, GFP, and BFP, which shows clear delivery and intracellular localization of a recombinant protein.

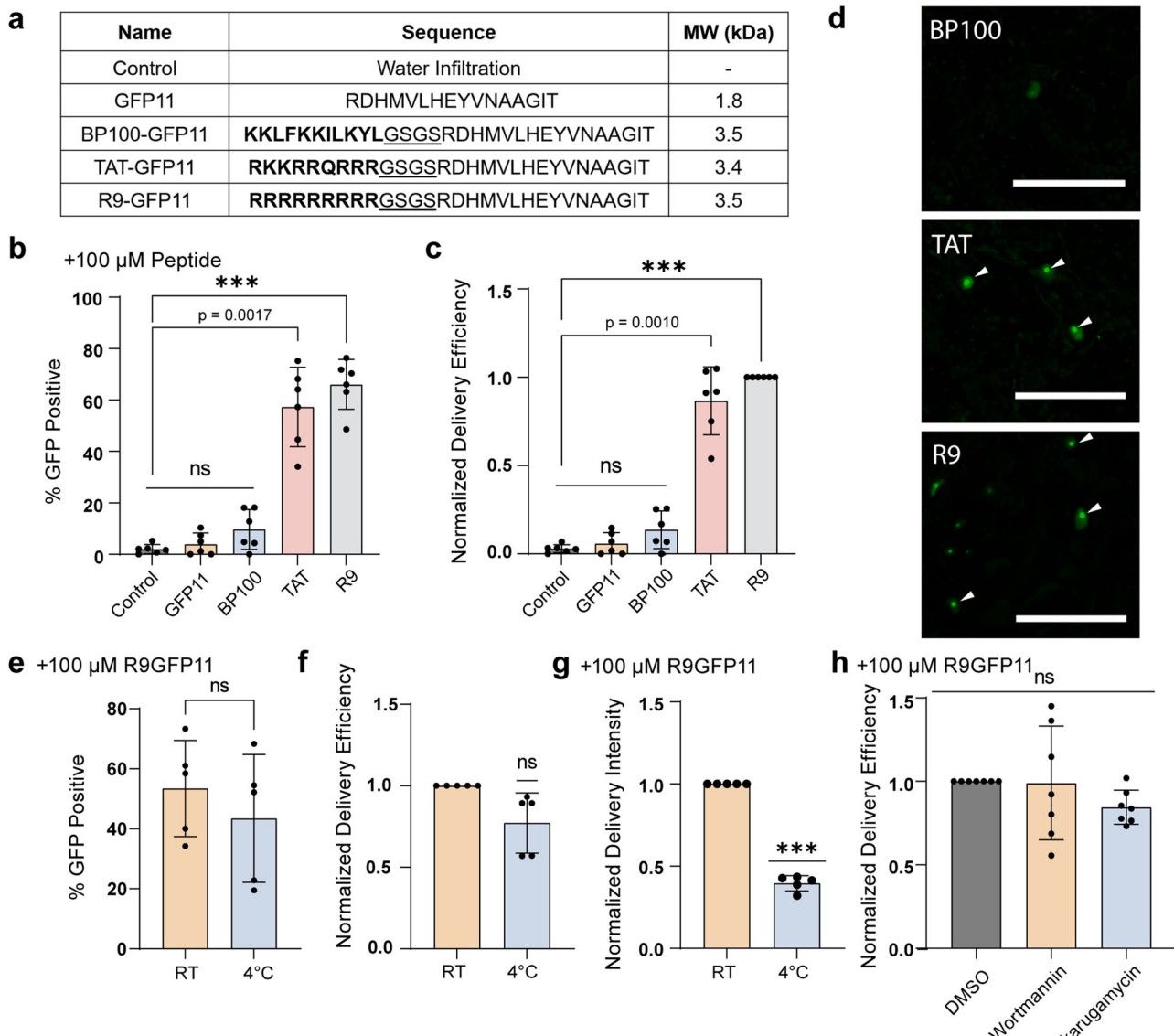

**Fig. 3 Investigating CPP performance using DCIP. a** Peptides tested for delivery of GFP11 and their corresponding sequence and molecular weights (kDa) with a water infiltration negative control. CPP sequences are bolded and the flexible GS linker is underlined. **b** Percent GFP-positive nuclei in leaf discs incubated with 100 μM with varying CPP-GFP11 conjugates for 4–5 h ($N = 6$) and, **c** delivery efficiency normalized to R9 CPP. **d** Representative micrographs of sfGFP fluorescent nuclei as the result of successful GFP11 delivery. White arrows point toward enhanced nucleolar localization of DCIP after delivery using arginine-rich CPP. **e** Average percent GFP positive in leaf discs treated with 100 μM R9-GFP11 and either left at room temperature (RT) or kept at 4 °C for 4–5H ($N = 5$) and, **f** corresponding delivery efficiency normalized to RT treatment. **g** Average normalized delivery intensity in leaf discs treated with 100 μM R9-GFP11 and either left at room temperature or kept at 4 °C for 4–5 h ($N = 5$). Delivery intensity is calculated by normalizing the mean green/red ratio of the 4 °C treatment to that of the RT treatment. For normalized results of the low-temperature treatment, a one-sample $t$-test comparing to the ideal value of 1.0 was used. **h** Normalized delivery efficiency in leaves infiltrated with 100 μM R9-GFP11 and co-infiltrated with either DMSO, 10 μM ikarugamycin, or 40 μM wortmannin for 4–5 h ($N = 7$). Efficiency is normalized to the DMSO group. Unless otherwise indicated, Kruskal–Wallis test followed by Dunn's multiple comparisons test was performed for all statistical comparisons where ns = $p > 0.05$, exact $p$-values are given for $0.001 < p < 0.05$, and ***$p < 0.001$. Scale bar is 100 μm. Error bars represent the standard deviation of the group repeats.

Owing to the low solubility of the protein and low frequency of delivery, automated quantification analysis resulted in no significantly quantifiable delivery (Supplemental Fig. S9c). A control treatment with GFP11-BFP expectedly did not result in observable or quantifiable delivery (Supplemental Fig. S9d).

**Delivery of f-actin-binding peptide with DCIP mediates protein–protein interactions in plants**. After confirming that recombinant proteins could be efficiently delivered into plant cells with R9 CPP, we asked if delivered proteins could subsequently mediate protein–protein interactions. Being able to mediate

protein–protein interactions using delivered proteins could enable new technologies that alter plant physiology. Therefore, in a proof-of-concept experiment, we delivered the 1.9 kDa f-actin binding peptide, Lifeact[50], tagged with GFP11 and R9 (Fig. 6a) into cytoDCIP expressing *N. benthamiana* leaves. In this scheme, when Lifeact is internalized, GFP11 acts as a scaffold to tether cytoDCIP to actin filaments through the Lifeact/f-Actin interaction. Kamiyama et al. had previously shown that GFP11–sfGFP1-10 interactions could be used to mediate protein scaffolding in mammalian cells[19]. Using our DCIP approach, we unambiguously confirm through imaging, that ectopic, delivery-mediated

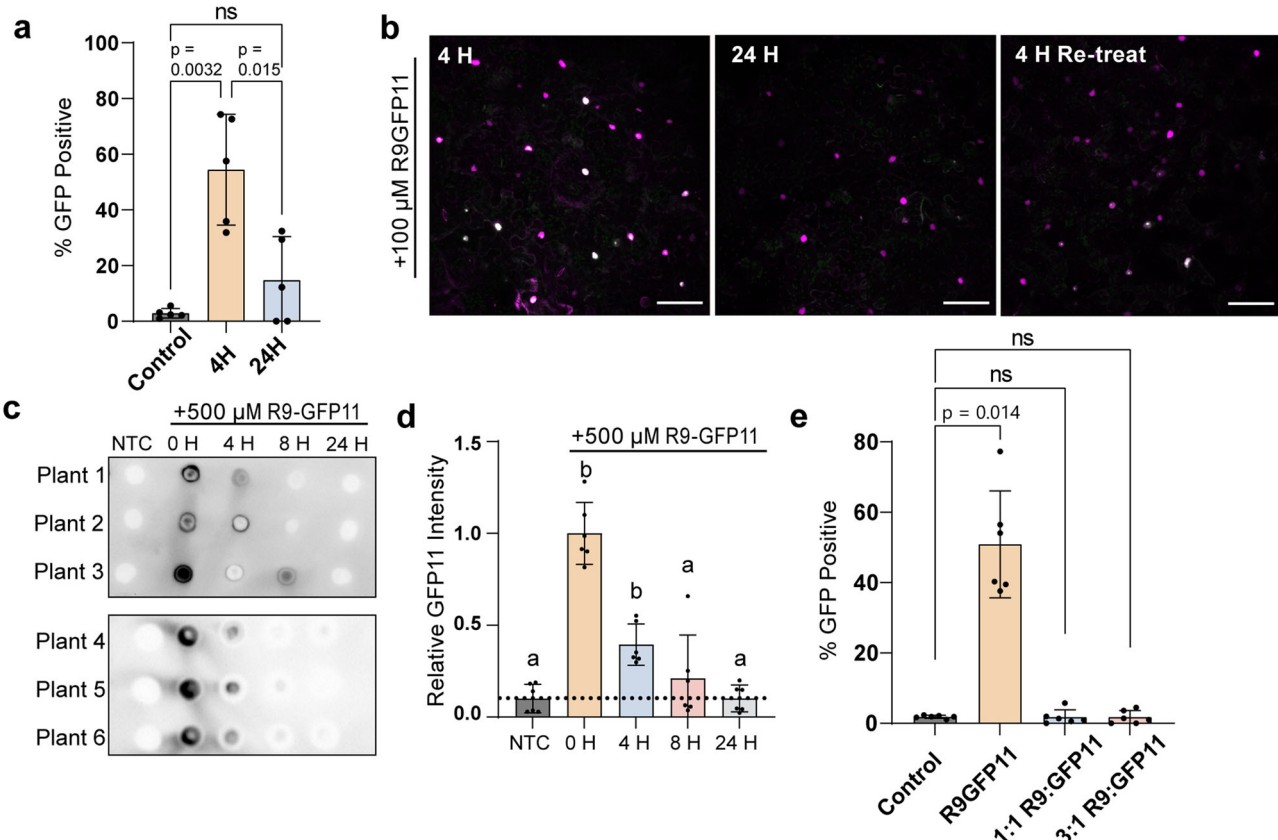

**Fig. 4 DCIP produces a transient response likely due to R9-GFP11 stability. a** Average percentage of GFP positive cells after infiltrating leaves with 100 µM R9-GFP11 for 4 or 24 h (N = 5). Kruskal–Wallis test followed by Dunn's multiple comparisons test was performed for all statistical comparisons where ns = p > 0.05, *0.01 < p < 0.05, and **p < 0.01. **b** Representative two-color confocal maximum intensity projection micrographs of intact leaves treated with 100 µM R9-GFP11 at 4, 24 h, or retreated for 4 h post 24 h treatment. mCherry nuclei are pseudocolored magenta and GFP green overlay results in white appearance. Scale bar is 100 µm. **c** Dot blot of lysates containing GFP11 recovered from leaves infiltrated with 500 µM R9-GFP11 or non-treated control for 0, 4, 8, and 24 h and immunoblotted with an anti-GFP11 antibody. **d** GFP11 spot intensity of non-treated leaves and leaves infiltrated with R9-GFP11 at 0, 4, 8, or 24 h were normalized to the mean intensity of the 0 h treatment to account for the variation in GFP11 recovery after lysis. Three separate plants were infiltrated for six biological replicates across two experiments (N = 6). Statistical comparison was performed with Kruskal–Wallis test where each letter represents groupings of statistical insignificance (p > 0.05). **e** Average percentage of GFP-positive cells at 4–5 h after infiltrating DCIP expressing leaves with 100 µM R9-GFP11, or 100 µM GFP11 with free R9 peptide in a 1:1 or 1:3 excess molar ratio. Kruskal–Wallis test followed by Dunn's multiple comparisons test was performed for all statistical comparisons where ns = p > 0.05 and exact p-values are given for 0.001 < p < 0.05. Error bars represent the standard deviation of the plotted points.

protein–protein interactions can be formed in plants. Previous studies using fluorescein-labeled BP100-Lifeact had shown that delivered dye-labeled Lifeact could bind to plant actin in BY-2 cells. However, the constructs were not tested in leaves and did not leverage the low background and scaffolding afforded by a bimolecular fluorescence complementation approach[51].

After a 6 h incubation with infiltrated 120 µM GFP11-Lifeact-R9, we observe robust labeling of actin filaments (Fig. 6b) in pavement cells. At higher zoom, numerous fine structures are observed as the result of Lifeact delivery (Fig. 6c). Also importantly, we see colocalization of the mCherry signal with the sfGFP signal in the filamentous structures (Fig. 6d), suggesting that the cytoDCIP has been scaffolded to the actin filaments. When cytoDCIP-expressing plants are treated with the molar equivalent of R9-GFP11, we observe no such fine structures and instead see diffuse, reticulated cytosolic localization (Fig. 6e). To further confirm that the observed filaments are indeed f-actin, we cotreated using either GFP11-Lifeact-R9 or R9-GFP11 with the f-actin-destabilizing drug, latrunculin B (LatB)[52,53]. In the presence of 25 µM LatB, no actin filaments were observed (Fig. 6f), which suggests depolymerization of f-actin and confirms that the structures imaged were indeed the result of mCherry-sfGFP1-10 scaffolding to f-actin. We did

however observe that LatB treatment changed the morphology of the cytosol in both the GFP11-Lifeact-R9 (Fig. 6f) cotreatment and the R9-GFP11 cotreatment (Supplemental Fig. S10). In both cases, the cytosol became granulated in a pattern with the appearance of numerous, cytosol-excluding compartments. Altogether, these data show that delivered proteins could be used to perturb protein localization and mediate protein–protein interactions *in planta* and that DCIP or cytoDCIP may help accelerate the identification of useful delivery-mediated protein–protein interactions.

**Recombinant WUSCHEL delivery assisted by DCIP analysis.**
After establishing the feasibility of using DCIP to assess recombinant protein delivery in intact leaves, we sought to examine R9-mediated delivery of the *Arabidopsis* plant morphogenic transcription factor, WUSCHEL (AtWUS), in *N. benthamiana* leaves. AtWUS was chosen as a candidate cargo due to its applications for somatic embryogenesis in plants and its high degree of molecular characterization[54,55]. DCIP-expressing leaves infiltrated with 140 µM GFP11-AtWUS-R9 (MW = 41 kDa) showed robust nuclear GFP complementation at 6H that does not

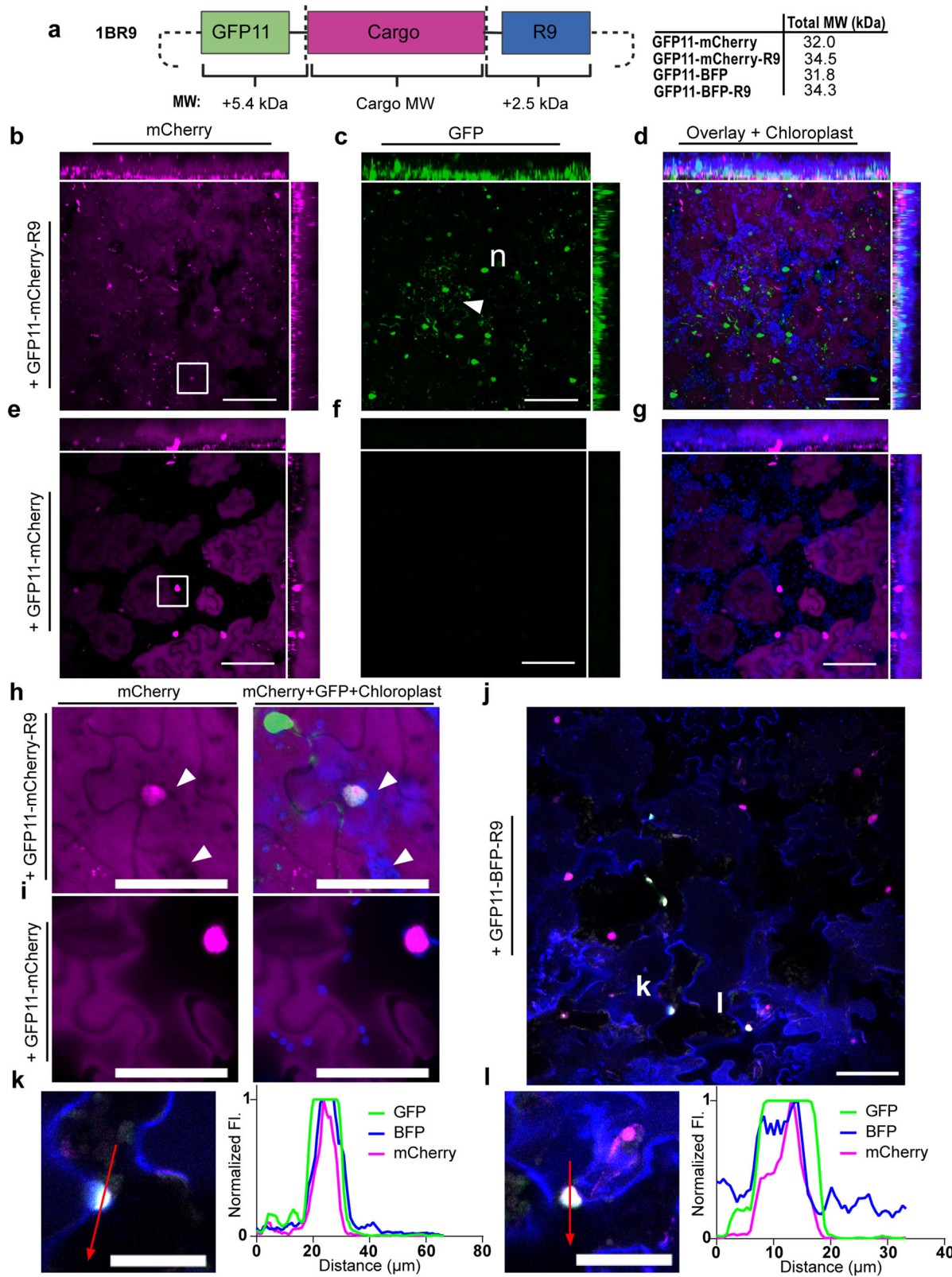

colocalize with plastid autofluorescence (Fig. 7a). Because AtWUS lacks intrinsic fluorescence unlike mCherry, we were also able to demonstrate quantitative DCIP analysis using GFP11-AtWUS-R9 (Fig. 7b) with 50% of DCIP-expressing cells being GFP positive, suggesting a 50% AtWUS delivery efficiency on a per-cell basis. Because AtWUS is a transcription factor, we also expected to see GFP11-AtWUS-R9 localize to the nucleus without the SV40 NLS

of DCIP. To test this, we infiltrated a cytoDCIP-expressing *N. benthamiana* leaf with 140 µM GFP11-AtWUS-R9. If the delivered AtWUS has active NLS activity, we would expect green fluorescence localized to only the nucleus and excess, uncomplemented mCherry-GFP1-10 to remain in the cytosol. Indeed, at 6 h we observe numerous GFP-positive nuclei surrounded by cytosolic mCherry fluorescence (Fig. 7c), thus confirming native

**Fig. 5 Qualitative confirmation of recombinant protein delivery. a** Schematic of LIC vector designed for tagging proteins of interest with N-terminal GFP11 and C-terminal R9 (created with BioRender.com). The resulting reporter and delivery tags result in an additional 7.9 kDa in MW to a recombinant protein of interest. **b–d** DCIP sensor expressing plant infiltrated with 60 μM GFP11-mCherry-R9 and incubated for 5 h. **b** mCherry fluorescence is pseudocolored magenta and demonstrates the ubiquitous presence of infiltrated GFP11-mCherry-R9. **c** sfGFP fluorescence from successful delivery is shown in green. Labeled features include nuclei (lowercase "n") and sfGFP containing intracellular aggregates or vesicular bodies (white triangle) resulting from successful delivery. **d** An overlay of mCherry and GFP channels as well as chloroplast autofluorescence (pseudocolored blue). **e–g** Equivalent experiment performed with 60 μM GFP11-mCherry showing, **e** mCherry fluorescence in magenta with some cells demonstrating strong DCIP nuclear fluorescence, **f** sfGFP fluorescence (green), and, **g** three color overlay with chloroplast channel. All images presented are maximum intensity projections with orthogonal projections of images. Scale bar is 100 μm. **h** Zoomed single-slice inset from panel **b** (white square) showing DCIP nuclear expression and colocalized GFP complementation as the result of successful delivery. White arrows mark dark voids in mCherry fluorescence caused by plastids in pavement cells that signify the intracellular presence of delivered mCherry. **i** Equivalent zoomed single-slice inset from panel **e** (white square) showing absence of plastid voids in mCherry signal. **j** representative single slice field of view from a DCIP expressing leaf treated with 100 μM total protein of GFP11-BFP-R9. mCherry is pseudocolored magenta, BFP is pseudocolored blue, and GFP is pseudocolored green. Overlap of all three colors as the result of delivered BFP and concomitant GFP complementation results in white pseudocolor. Scale bar is 50 μm. White "k" and "l" mark nuclei used in subsequent profile plots. **k** and **l** representative profile plots of nuclei with successful BFP delivery and GFP complementation. Fluorescence intensities are normalized to the maximum in the profile. For inset images, scale bar is 50 μm.

NLS targeting of delivered AtWUS. In contrast, R9-GFP11-treated cytoDCIP shows a general, cytosolic localization (Fig. 6e). These data show that R9 fusion is effective for WUS delivery and that the purified R9-tagged transcription factor is able to enter the nucleus.

The previous results in *N. benthamiana* motivated us to determine whether or not the delivered AtWUS is transcriptionally active. We treated 12-day-old *Arabidopsis* seedlings for 24 h with 1 μM GFP11-AtWUS-R9. *A. thaliana* was chosen as a model species due to the well-characterized AtWUS pathway in *A. thaliana*. Seedlings were subsequently harvested and subjected to RT-qPCR analysis for six known direct targets of AtWUS (Fig. 7d). GFP11-AtWUS-R9 delivery led to down-regulation of *ARR6* (0.42-fold), *AS2* (0.36-fold), *KAN1* (0.58-fold), *KAN2* (0.44-fold), *YAB3* (0.28-fold), and upregulation of *CLV3* (2.06-fold), when compared to the buffer-treated control. Statistical analysis of the measured $C_T$ values shows that R9-GFP11 alone does not mediate significant transcriptional changes while GFP11-AtWUS-R9 recapitulates the expected transcriptional response to AtWUS overexpression (Fig. 7e). Ectopic expression of AtWUS downregulates the expression of cell identity markers *ARR6*, AS2, *KAN1/2* and *YAB3* by binding directly to their promoters[56,57], and upregulates the expression of its own negative regulator, *CLV3*[58]. We repeated the experiment with a separately purified batch of proteins and observed similar results with all expected down-regulated targets being downregulated, although *CLV3* was not upregulated to a statistically significant degree and *ARR6* was only down-regulated relative to the R9-GFP11 control (Supplemental Fig. S11). In this second experiment, 3 μM of protein was also used due to apparent batch-to-batch variation of activity and lower purity of the recombinant protein (Supplemental Fig. S12b, c). These data suggest that not only is AtWUS delivery possible but that this delivered transcription factor can be transcriptionally active in plants.

We also investigated whether GFP11-AtWUS on its own is cell penetrating in contrast to either mCherry or BFP. To our surprise, AtWUS was found to enter plant cells without R9 fusion (Fig. 7f) with similar efficiency as the R9-containing construct at 140 μM (Fig. 7g). The protein also possessed nuclear localization activity when infiltrated into a cytoDCIP-expressing leaf (Supplemental Fig. S13a). In search for an explanation for the native cell-penetrating behavior, we aligned the third homeodomain helix of AtWUS with several animal homeodomain proteins (ANTENNAPEDIA, VAX1, OCT4) (Fig. 7h) which have been also shown to be cell penetrating in mammalian cells[59–61] and two plant homeodomain proteins (WUS2 and STM).

Moreover, a global analysis of homeodomain proteins in human cells shows the majority to be cell-penetrating and potential paracrine signaling molecules[62]. Furthermore, one of the first CPPs, characterized, penetratin (RQIKIWFQNRRMKWKK), is derived from the conserved homeodomain helix of *Drosophila* ANTENNAPEDIA and the conserved homeodomain of *Maize* KNOTTED-LIKE 1 is reported as cell-penetrating in mammalian cells[59,61]. To further confirm this, we synthesized a peptide fragment consisting of the conserved amino acids 82–102 (KNVFYWFQNHKARERQKKRFN) of AtWUS fused to GFP11 (WUSP-GFP11) (Fig. 7h). Using confocal analysis, we found that this peptide was indeed cell penetrating (Supplemental Fig. S13b), although not as effective as R9 CPP (Fig. 7i). Deletion of amino acids 82–102 from GFP11-AtWUS (Δα3) resulted in a protein with compromised cell penetration ability (Δα3 = 15.7% vs. WT = 51% mean GFP positive) and treatment of DCIP-expressing plants with this protein resulted in non-significant ($p > 0.05$) delivery regardless of an outlier test-based exclusion of a repeat where high-level uptake was measured (Supplemental Fig. S13c, d). The low, basal-level delivery of Δα3 that occurs may be the result of low amounts of non-specific uptake at the investigated concentration. Although these results may seemingly contradict previous research showing plasmodesmata-based trafficking of WUS[63], our direct, exogenous delivery method bypasses the plasmodesmata and relies on diffusion through the cell wall and penetration through the plasma membrane. GFP11-AtWUS protein also showed lower transcriptional activity than the R9 CPP-containing construct, only regulating half (3/6) or one-sixth (1/6) of the tested genes in an expected manner across two experimental repeats respectively (Supplemental Fig. S13e–h). Unexpectedly, we also observed CLV3 downregulation in GFP11-AtWUS-treated plants. One possible explanation is the AtWUS concentration-dependent regulation of the WUS-CLV3 axis[64]. The observed lower delivery efficacy of just the AtWUS-derived CPP may also explain the difference in transcription-modulating activity between the GFP11-AtWUS-R9 construct and the GFP11-AtWUS at low, micro-molar concentrations.

## Discussion

Protein delivery in plants is an emerging field motivated by the need for better handles to dissect plant molecular physiology, enable DNA-free gene editing technology, and enhance agronomic traits. Previous foundational work by Numata et al. has identified cell-penetrating peptides as a method to deliver proteins using a dye-mediated approach[24]. However, the journey from the apoplast into the cytoplasm causes cargoes to be excluded, entrained, or sequestered and necessitates a cargo-in-

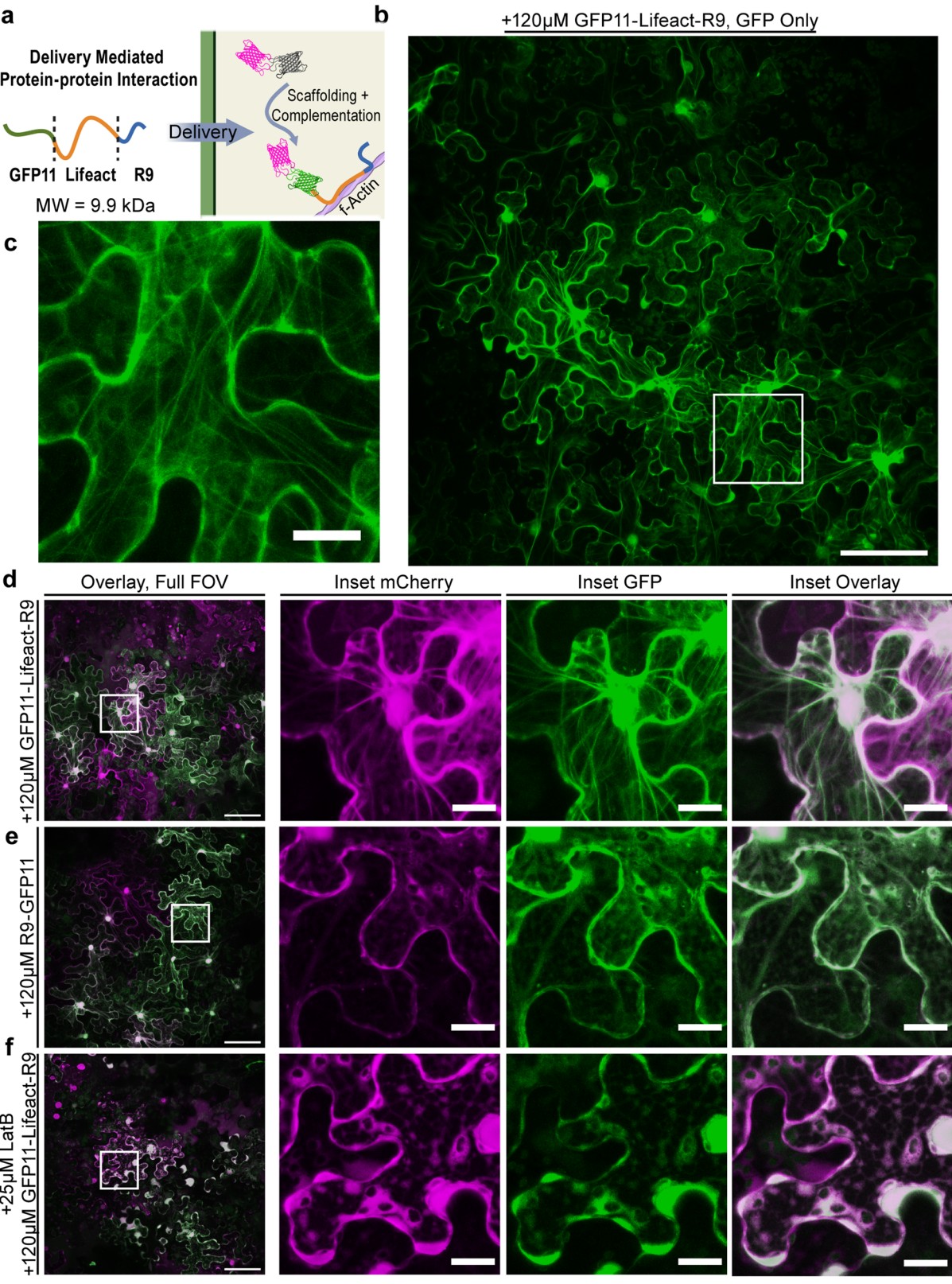

cytosol-dependent approach. Furthermore, technical limitations render microscopic analysis of uptake only suggestive of uptake rather than *confirmative* and preclude quantification of relative CPP delivery efficiencies. Previously, final confirmation of delivery in plants has required lengthy functional analysis such as identifying delivery-mediated gene editing[65], protein expression[25,66], or silencing[7]. We, therefore, sought to develop a

technique by which delivery designs could be rapidly and accurately assessed, shown here for CPP-mediated peptide and protein delivery but could be generically extended to testing of other carriers. Although a suite of tools has been developed for use in mammalian cell culture to confirm CPP-mediated cytosolic delivery[21,22,67], no such tool has been tailored for plants until the present work.

**Fig. 6 F-actin delivery-mediated protein–protein interactions formed in plants. a** Schematic of delivery-mediated protein–protein interactions (created with BioRender.com) using purified recombinantly expressed GFP11-Lifeact-R9 (9.9 kDa). Infiltrated GFP11-Lifeact-R9 enters the cytosol through CPP-mediated delivery, binds to f-actin filaments in the plant cell, and scaffolds mCherry-sfGFP1-10 to the actin filament, thus mediating protein–protein interactions. **b** Representative standard deviation projection FOV of actin labeling enabled by cytoDCIP and delivered GFP11-Lifeact-R9. CytoDCIP-expressing leaves were infiltrated with 120 μM GFP11-Lifeact-R9 and incubated for 6 h as leaf discs. **c** Inset (white square) showing fine filament detail. **d** Filaments appear as sfGFP fluorescent structured strands that colocalize with mCherry as a result of cytoDCIP scaffolding. **e** control treatment of CytoDCIP expressing leaves using 120 μM R9-GFP11 shows diffuse cytosolic localization of both mCherry and sfGFP. **f** GFP11-Lifeact-R9 cotreated with 25 μM Latrunculin B results in actin depolymerization in diffuse cytosolic staining only. In all images, mCherry is pseudocolored magenta and sfGFP is pseudocolored green. Full FOV scale bar is 100 μm and the inset 20 μm.

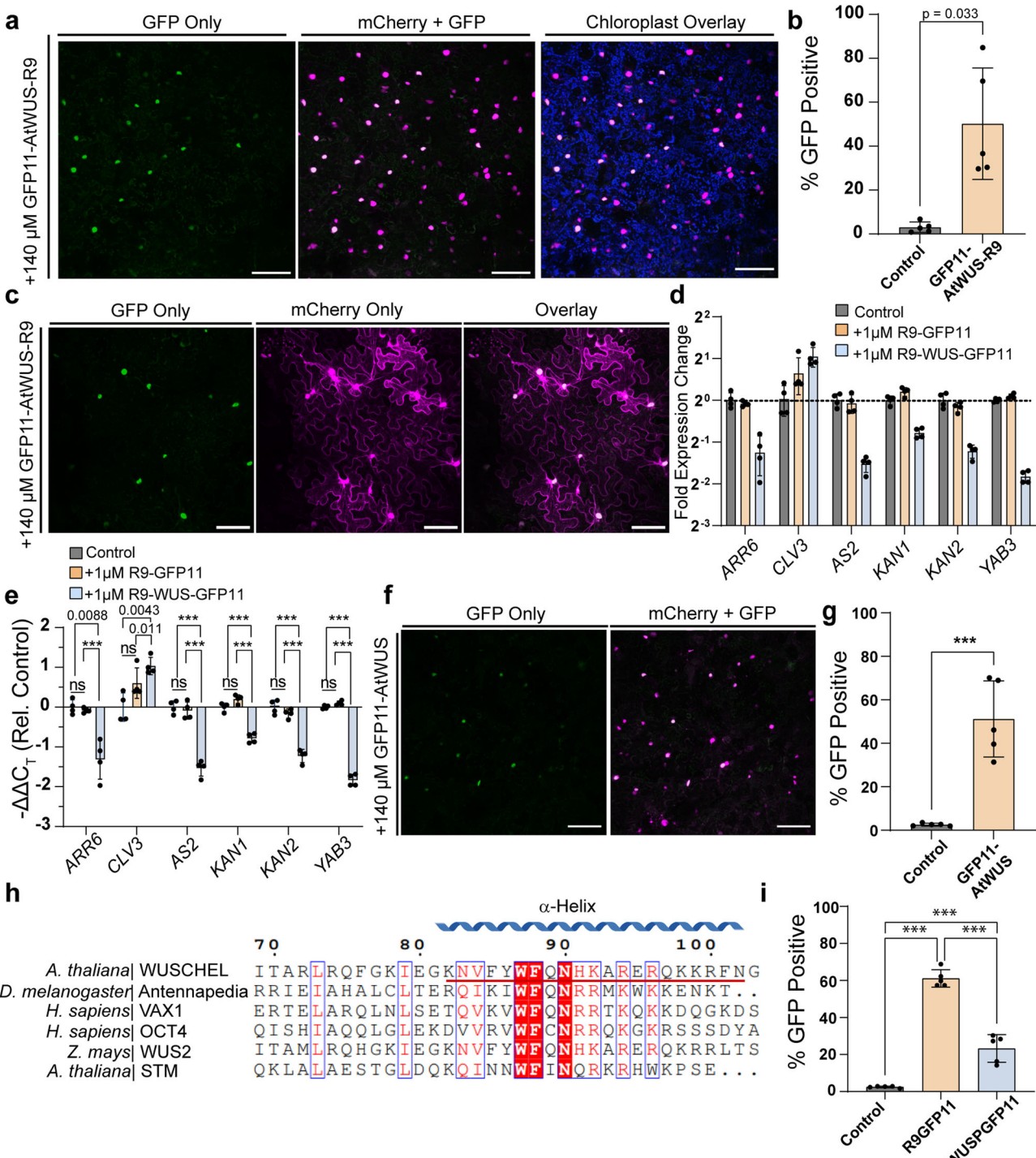

**Fig. 7 Delivery of the morphogenic transcription factor WUSCHEL. a** Representative maximum intensity projection of a DCIP-expressing *N. benthamiana* leaf infiltrated with 140 μM GFP11-AtWUS-R9 and incubated as a leaf disc for 6 h. Successful delivery presents as sfGFP (green pseudocolor) and mCherry (magenta pseudocolor) fluorescent nuclei as the result of delivered complementation. **b** Quantification of GFP-positive nuclei as a result of AtWUS delivery in five plants ($N = 5$) or buffer control. **c** Representative maximum intensity projection of cytoDCIP expressing *N. benthamiana* leaf infiltrated with 140 μM GFP11-AtWUS-R9 and incubated as a leaf disc for 6 h. The exclusive localization of sfGFP fluorescence and ubiquitous localization of cytosol-localized cytoDCIP mCherry fluorescence shows that delivered GFP11-AtWUS-R9 is able to undergo native nuclear localization. **d** rt-qPCR analysis of downstream AtWUS genes in 12-day-old *Arabidopsis* seedlings treated with 1 μM GFP11-AtWUS-R9 or R9-GFP11 for 24 h. 8–10 seedlings were treated per well and four wells were utilized for each treatment ($N = 4$). **e** Statistical comparison showing measured $\Delta\Delta C_T$ values of GFP11-AtWUS-R9-treated seedlings are significantly changed when compared to either R9-GFP11 treatment or no treatment. Statistical analysis performed with *t*-test comparison and Holm–Šídák correction for multiple comparisons where ns = $p > 0.05$, *$0.01 < p < 0.05$, **$p < 0.01$, ***$p < 0.005$. **f** Representative maximum intensity projection of a DCIP expressing *N. benthamiana* leaf infiltrated with 140 μM GFP11-AtWUS and incubated as a leaf disc for 6 h. Successful delivery presents as sfGFP (green pseudocolor) and mCherry (magenta pseudocolor) fluorescent nuclei as the result of delivered complementation. **g** Quantification of GFP positive nuclei as a result of AtWUS delivery in five plants ($N = 5$) or buffer control. **h** Sequence alignment of AtWUS with several other plant and animal homeodomain transcription factors centered around putative conserved cell penetrating helix using Clustal Omega and ESPRIPT. Conserved residues are highlighted in red, while similarly charged residues are boxed in blue. The tested AtWUS-derived CPP (WUSP) is underlined in red. **i** Quantitative microscopy DCIP results for *N. benthamiana* treated with either 100 μM R9-GFP11 or 100 μM WUSP-GFP11 and imaged at 4–5 h post infiltration. Statistical analysis was performed with *t*-test comparison and Holm–Šídák correction for multiple comparisons where ns = $p > 0.05$, exact *p*-values are given for $0.001 < p < 0.05$, and ***$p < 0.001$. All scale bars are 100 μm. All error bars represent the standard deviation of the plotted points.

Our development, delivered complementation *in planta* (DCIP) is a microscopic tool that unambiguously confirms the delivery of peptides and proteins even at concentrations as low as 10 μM in a complex tissue such as leaves. DCIP also enables quantitative measurement of relative delivery efficiency, thus enabling a new functional method to rapidly screen effective cell-penetrating peptides. In our assay, we identified TAT and R9 as being effective *in planta*-delivery CPPs, with R9 as the most effective, whereas the previously identified BP100[24] was ineffective at delivering GFP11. One potential explanation for this discrepancy is the sensitivity of CPPs to chemical conjugation and that dye-conjugation as used in previous CPP screens perturbs their uptake performance[68]. Furthermore, the uptake of CPP-conjugated cargoes is dependent on the chemical properties of the cargo as well[69]. The use of GFP11 in DCIP as a reporter tag presents an improvement to screening strategies as it provides a clear, low-background signal of successful cytosolic delivery that is bio-similar to protein cargoes. We additionally used DCIP to show that the uptake of R9 in plants is endocytosis-independent, in line with previous literature in mammalian systems[33]. Unraveling the CPP uptake mechanism in plants may lead to novel strategies for improving peptide and protein delivery *in planta*. These experiments show DCIP is a tool that can be used both to interrogate delivery and also to screen novel CPP sequences and chemistries that could improve the delivery of not just proteins but eventually RNA or DNA.

DCIP has also enabled us to study the durability of delivered peptides *in planta*. We used DCIP and observed that the sfGFP complementation response from R9-GFP11 delivery was transient and disappeared by 24 h. We orthogonally confirmed that R9-GFP11 is cleared from leaf tissue by 24 h with immunoblotting and show that R9 conjugation to GFP11 is required for successful delivery These results suggest that the stability of cargo and conjugation of a cargo to a cell-penetrating motif will be an important consideration for designing effective plant bioengineering strategies that leverage delivery, and that successful delivery is also contingent on cargo stability in addition to delivery itself. These results align with previous nanoparticle-mediated delivery strategies that involve stabilization of the cargo to lytic enzymes but not necessarily cellular internalization of the vehicle itself[7]. DCIP, therefore, offers a facile tool for engineering novel carrier-based designs that optimize both delivery efficiency and cargo stability.

Next, we used DCIP and a cytosolically localized variant, cytoDCIP, to investigate the delivery of larger recombinant proteins. We show that both GFP11-mCherry and GFP11-BFP can be delivered using a c-terminally fused R9 CPP to leaf cells. In contrast to previous recombinant fluorescent protein delivery strategies, the bimolecular fluorescence complementation signal we observe can only occur upon successful cytosolic delivery and is not convolved with apoplastic or endosomally entrapped material. As a proof of concept that future experiments that leverage delivery-mediated protein–protein interactions may be possible, we then used cytoDCIP to show that specific protein–protein interactions can be mediated through a delivered recombinant protein by tethering the mCherry-sfGFP1-10 to f-actin through Lifeact/f-actin binding. Incidentally, the development of GFP11-Lifeact-R9 may also provide a novel way to visualize actin filaments in plants.

As a final demonstration of the usefulness of DCIP, we show, to our knowledge, the first delivery of a plant transcription factor, AtWUS, to walled plant cells. We not only show, through imaging, that it is possible to deliver recombinant AtWUS to *N. benthamiana* but also show, through RT-qPCR, that delivered AtWUS recapitulates AtWUS overexpression transcriptional downstream responses in *Arabidopsis* seedlings. DNA-based overexpression of AtWUS and its orthologs has been found to enhance the regeneration of transgenics and somatic embryogenesis of numerous species such as cotton, sorghum, and maize[3,70,71]. The confirmation of active AtWUS delivery evinces a new DNA-free strategy for enhancing the recovery of transgenic plants without DNA-based WUS overexpression and subsequent transgene excision in challenging species. However, future work will be needed to develop a strategy for delivered WUSCHEL-mediated somatic embryogenesis. In an unexpected turn, we used DCIP to discover a new cell-penetrating peptide in plants and provide evidence that plants homeodomain proteins may be generally cell-penetrating, broadening the list of potential, readily deliverable plant transcription factors. For example, the morphogenic regulators *Zea mays* WUS2 and *Arabidopsis thaliana* STM[3] show high sequence similarity with other cell-penetrating homeodomains (Fig. 7h).

In summary, our development and validation of DCIP, *Delivered Complementation* in *planta*, could enable the design of novel cell-penetrating peptides or nanoparticle-based carriers for protein delivery. For example, DCIP might be used to identify plant-specific cell penetrating motifs from secreted effector proteins of pathogenic fungi[72]. DCIP may also be used in future experiments to determine whether larger proteins such as Cas9 or novel miniaturized Cas12f variants could be delivered[73]. In

addition to the aforementioned DNA-free gene editing, DCIP could engender the delivery of nanobodies for pathogen resistance and targeted protein degradation[74,75], the delivery of stress tolerance conferring disordered proteins[76], or the delivery of a greater variety of morphogenic regulators to control plant regeneration.

## Methods

**Reagents and antibodies**. Reagents, buffers, and media components were procured through Sigma-Aldrich unless otherwise noted. Solid-phase chemical peptide synthesis of GFP11 and CPP fusions was performed by a third-party manufacturer (GenScript). Enzymes used for cloning reactions were procured through New England Biolabs. Anti-GFP11 antibody was purchased through Thermo-Fisher and the anti-mCherry and anti-rabbit Igg-HRP secondary antibody through Cell Signaling Technologies. All oligonucleotides and DNA sequences were purchased from Integrated DNA Technologies (IDT).

**Plant growth conditions and agroinfiltration**. *N. benthamiana* were grown in a growth chamber kept at 24 °C and a light intensity of 100–150 μmol m$^{-2}$ s$^{-1}$. The photoperiod was kept at 16 h light/8 h dark. Seeds were sown in inundated soil (Sunshine Mix #4) and left to germinate for 7–10 days at 24 °C before being transferred to 10 cm pots for growth. Fertilization was done on a weekly basis with 75 ppm N 20-20-20 general-purpose fertilizer and 90 ppm N calcium nitrate fertilizer reconstituted in water. Infiltrations were performed on 4–5-week-old plants on the third and fourth expanded leaves. Agroinfiltrations were performed via needless syringes using overnight cultures of *A. tumefaciens* bearing the DCIP constructs. On the day of infiltration, the overnight 30 °C cultures were pelleted at 3200 × *g*, rinsed with infiltration buffer (10 mM MES pH 5.7, 10 mM MgCl$_2$), and then resuspended in infiltration buffer containing 200 μM acetosyringone to an OD of 0.5–1.0. The cultures were then left shaking at ambient conditions for 2–4 h before the final adjustment of the OD$_{600}$ to 0.5 with infiltration buffer. During infiltration, care was taken to minimize the number of damaging infiltration spots to completely saturate the leaf. The plants were then left at ambient conditions overnight to dry before being transferred to the growth chamber for a total incubation time of 3 days.

**Plasmid construction and bacterial strains**. A list of parent plasmids and newly constructed plasmids are included in Supplemental Data 1. Additionally, the predicted protein products and organization of all constructed plasmids are provided. Primers and synthetic DNA used for cloning are included in Supplemental Data 2. For all cloning steps, plasmids were transformed into XL1-blue *E. coli*. The coding sequence of DCIP was constructed by ligating the mCherry sequence into a PstI 5′ of the sfGFP1-10 coding sequence in pPEP101[77]. In the nuclear-localized variant of DCIP, NLS was attached during PCR of the mCherry sequence. The NLS is omitted in cytoDCIP. Next, the coding sequences of DCIP and cytoDCIP were amplified by PCR for domestication into pUDP2 before the final Golden Braid (GB2.0) assembly following the standard GB2.0 protocol using Esp3I[28]. The DCIP and cytoDCIP transcriptional units were assembled using pUDP2-35S-oTMV for the promoter and pUDP2-tNOS for the terminator in a BsaI restriction-ligation GB2.0 reaction. cytoDCIP and DCIP were then transformed into GV3101 *Agrobacterium tumefasciens* bearing pSOUP[78] and plated onto LB agar containing rifampicin (50 μg/mL), gentamicin (25 μg/mL), and kanamycin (50 μg/mL).

The recombinant GFP1-10 expression vector, 1B-GFP1-10, was constructed by PCR amplifying the sfGFP1-10 gene from a pPEP101 with the ligation-independent cloning tags for plasmid 1B. The full protocol for LIC cloning used was provided by the UC Berkeley Macro Lab: https://qb3.berkeley.edu/facility/qb3-macrolab/projects/lic-cloning-protocol/. The resulting amplicon was then inserted by LIC into 1B and transformed into *E. coli* for expansion, purification, sequencing, and transformation into an expression *E. coli* strain. The 1BR9 plasmid was constructed by inserting a short, chemically synthesized DNA sequence containing N-terminal GFP11 and C-terminal R9 tag into plasmid 1B via LIC. Between the tags, a new LIC site was regenerated such that future LIC reactions would insert the protein of interest between the N- and C-tags. The LIC approach allowed the insertion of PCR-amplified mCherry and BFP sequence into 1BR9 to generate 1BR9-mCherry and 1BR9-BFP. Incorporation of a TAA stop codon into the reverse primer generated 1BR9-mCherrySTOP and 1BR9-BFPSTOP which excludes the c-terminal R9 motif. 1BR9-Lifeact was produced by inserting a chemically synthesized DNA sequence for Lifeact into 1BR9. 1BR9-AtWUS and 1BR9-AtWUSSTOP were constructed by LIC insertion of an *E. coli* codon-optimized (IDT) synthesized DNA AtWUS (TAIR: At2g17950.1) DNA sequence with or without a TAA stop codon into the 1BR9 vector. 1BR9-AtWUSSTOP-Δα3, for the expression of GFP11-AtWUS- Δα3, was created using around-the-horn cloning with 5′ phosphorylated primers to exclude the third alpha helix of the homeodomain via PCR using 1BR9-WUSSTOP as the template.

**Recombinant protein expression and purification**. For all recombinant protein expression, Rosetta 2 (DE3) pLysS *E. coli* were transformed with recombinant protein expression vectors and plated onto selective chloramphenicol (25 μg/mL)

and kanamycin (50 μg/mL) LB agar plates for overnight growth at 37 °C, 250 rpm. Single colonies were used to inoculate 10 mL seed cultures in LB. After overnight starter culture growth at 37 °C and 250 rpm, 1 L LB with selective antibiotics was set to grow at 37 °C, 250 rpm in 2 L baffled flasks. Induction was performed with 0.5 mM IPTG at 37 °C when the culture reached 0.8 OD$_{600}$. After 4 h of induction, cultures were pelleted for 20 min at 3200 × *g* and flash-frozen in liquid nitrogen. Cell lysis was conducted using thawed pellets in lysis buffer (50 mM Tris–HCl, 10 mM imidazole, 500 mM NaCl, pH 8.0) with 1x protease inhibitor cocktail (Sigma-Aldrich: S8830) using probe tip sonication. The resulting lysate was clarified by centrifugation at 40,000 × *g* for 30 min. For GFP1-10 and mCherry fusion proteins, the soluble fraction was incubated with 1 mL Ni-NTA (Thermo Scientific: 88221) slurry for an hour. After incubation and washing, the proteins were eluted using elution buffer (500 mM imidazole, 150 mM Tris–HCl, pH 8.0). The resulting eluate was then concentrated and buffer exchanged into 10 mM Tris–HCl, 100 mM NaCl pH 7.4 via ultrafiltration in a 3500 Da cutoff filter (Emdmillipore: C7715). The GFP1-mCherry and GFP11-mCherry-R9 were then further polished using SEC (Cytiva: HiLoad 16/600 Superdex 200 pg) and exchanged into storage buffer (10 mM Tris, 10 mM NaCl pH 7.4) before ultrafiltration concentration and flash freezing for storage. BFP purification proceeded similarly except for the R9 construct where all steps are done at pH 10.8, 20 mM CAPS buffer instead of Tris. For GFP11-Lifeact-R9, the insoluble pellet from clarification was solubilized in 8 M Urea, 50 mM Tris, pH 8.0 before incubation with Ni-NTA for an hour. In addition to standard wash steps, a high pH wash (20 mM CAPS pH 10.8, 1 M NaCl) was required to remove residual nucleic acids from the protein. Protein was then eluted with elution buffer before spin concentration and exchanged into storage buffer and flash freezing. Aliquots of each recombinant protein were run on SDS–PAGE for confirmation (Supplemental Figs. S9a and S12a).

**Recombinant GFP11-AtWUS purification**. GFP11-AtWUS-R9, GFP11-AtWUS, and GFP11-AtWUS-Δα3 were expressed as mentioned previously. However, purification proceeded by sonication lysis in 6 M Urea, 50 mM Tris, 0.5 mM TCEP, and 2 mM MgCl$_2$ pH 7.5 in the presence of 25 U/mL of benzonase and protease inhibitor cocktail. After lysis, the lysate was then incubated at 37 °C for 30 min with occasional mixing. After incubation, an additional 25 U/mL of benzonase was added and the lysate was clarified by centrifugation at room temperature, 40,000 × *g* for 30 min. To the clarified supernatant, 2 mL of Ni-NTA slurry per liter of starting culture was added and incubated at room temperature for 2 h. The Ni-NTA was then washed sequentially with at least 15-bed volumes each of wash A (6 M Urea, 50 mM Tris pH 7.5), then wash B (6 M Urea, 50 mM Tris, pH 7.5, 500 mM NaCl), then wash C (6 M Urea, 50 mM Tris pH 7.5, 50 mM Imidazole). The protein was eluted twice with the addition of 1-bed volume of 1 M Imidazole, 20 mM MES, pH 6.9, and 200 mM NaCl. The fractions were then combined and buffer exchanged into buffer P (30 mM sodium acetate, 5 mM NaCl, pH 4.0, 0.5 mM TCEP) by dialysis or by desalting in a G25 Sephadex pre-packed PD-10 column (Cytiva) and then analyzed with SDS–PAGE (Supplemental Fig. S12b–d) and in vitro complementation (Supplemental Fig. S3c–e).

**In vitro GFP complementation assay**. In a 96-well qPCR plate (Biorad), 5 μL of 10 μM total protein of sfGFP1-10 Ni-NTA eluate in storage buffer (10 mM Tris, 10 mM NaCl pH 7.4) was combined with 5 μL buffer or 20 μM of each tested peptide or protein in storage buffer. Each well was mixed by pipetting and was tested in triplicate. GFP complementation of AtWUS proteins was accomplished similarly except 10 μL of 15 μM either GFP11-AtWUS-R9 or R9-GFP11 in buffer P was added to 10 μL of 10 μM GFP1-10. An additional 5 μL 1 M Tris pH 8.0 was also added to overcome the acidity of buffer P. GFP complementation was quantified using a Biorad CFX96 qPCR machine by measuring the green fluorescence at 1-min intervals over the course of 6 h at either 22 or 4 °C.

**SDS–PAGE and Western Blot**. For the Western Blot of DCIP and cytoDCIP, agroinfiltrated leaves were harvested at 3 d.p.i, flash frozen, ground, and lysed with RIPA buffer (Abcam: ab156034) for 20 min. Lysates were then clarified by centrifugation and boiled in 1X Laemmli buffer (Biorad) and 10 (V/V) % beta-mercaptoethanol for 5 min before being loaded into a 4–20% gradient SDS–PAGE gel (Biorad: 4561096) and run according to the manufacturer's instructions. Membrane transfer was done according to the manufacturer's instructions onto Immobilon PVDF membrane (Millipore). Blocking was performed using 5% milk in PBS with 0.1% Tween (PBST). The incubation with primary anti-mCherry antibody (CST: E5D8F) was performed overnight at 1:1000 dilution in 3% BSA in PBST at 4 °C with orbital shaking at 60 rpm. Imaging was performed after probing with anti-Rabbit IGG-HRP secondary antibody (CST: 7074) at 1:10,000 dilution in 5% milk PBST and ECL prime chemiluminescent reagent (Amersham: RPN2236) on a ChemiDoc gel imager (Biorad).

**R9-GFP11 dot blot assay**. The third or fourth leaf of 4–5-week-old wild-type *N. benthamiana* was infiltrated with 500 μM R9-GFP11 in water using a needless syringe. Infiltrations were staggered such that all treatments could be harvested simultaneously for 0, 4, 8, and 24 h time points. Each treatment was performed on a separate plant and the experiment was repeated thrice. A 12 mm leaf disc was excised for each treatment using a leaf punch and flash frozen in liquid nitrogen

before grinding and lysis in 20 μL RIPA buffer with 1x plant protease inhibitor cocktail (Sigma-Aldrich: P9599). The lysates were then clarified by centrifugation at 21,000 × *g* for 30 min. Immediately after, 2 μL lysates were spotted onto a nitrocellulose membrane (Amersham: GE10600002) and allowed to dry. The membranes were then blocked in 5% milk in PBS-T before being washed and probed with an anti-GFP11 antibody (Invitrogen: PA5-109258) (1:500 dilution in PBST with 3% BSA). Secondary antibody probing and imaging were performed similarly to Western Blot.

**Delivered complementation *in planta* infiltration**. Three days after agroinfiltration, DCIP-expressing leaves were infiltrated with the treatment solutions. Unless otherwise stated, an 8 mm punch was then excised from the infiltrated area and plated, abaxial side up, onto ½ MS pH 5.7 agar plates. The plates were then left to incubate under ambient conditions for 4–5 h before imaging. For cold temperature treatment, after infiltration with ice-cold solutions of R9-GFP11 and disc excision, the leaf discs were plated onto ice-cold agar and immediately transferred to a 4 °C refrigerator for incubation. After incubation, the agar plates with leaf discs were kept on ice until the moment of imaging. For in situ incubation, leaves were simply infiltrated and the plant was returned to the growing chamber for incubation. All peptides were dissolved in sterile water and all recombinant proteins were dissolved in 10 mM Tris pH 7.4, 10 mM NaCl for infiltration. GFP11-AtWUS-R9 was found to possess low solubility at neutral pH and was exchanged into 10 mM MES, pH 5.5, immediately before infiltration.

**Confocal imaging and image analysis**. Excised leaf discs were imaged on a Zeiss LSM880 laser scanning confocal microscope. Images for semi-quantitative analysis were collected using a ×20/1.0NA Plan-apochromat water immersion objective and larger field-of-views were collected using a ×5 objective. Leaf discs were mounted by sandwiching a droplet of water between the leaf disc and a #1.5 cover glass. BFP, sfGFP, mCherry, and chloroplast autofluorescence images were acquired by excitation with a 405, 488, 561, and 635 nm laser, respectively. The emission bands collected for BFP, sfGFP, mCherry, and autofluorescence were 410–529, 493–550, 578–645, and 652–728 nm, respectively. All images were collected such that the aperture was set to 1 Airy-unit in the mCherry channel. Images and profile plots were prepared for publication using Zen Blue software. Profile plots were smoothed by taking a moving window of three measurements and normalized to the maximum smoothed intensity for each color. For quantification experiments, z-stacks were acquired with the imaging depth set to capture the epidermal layer down to the point where mCherry nuclei could no longer be detected. z-stacks from four fields of view were acquired for every treatment condition. Quantitative image analysis and downstream processing was performed using Cell Profiler 3.0. On a slice-by-slice basis, mCherry fluorescent nuclei were segmented using Otsu's method[79] and chloroplasts were segmented in the autofluorescence channel. The segmented chloroplasts were applied as a sfGFP channel mask over the image to exclude plastid autofluorescence in downstream image analysis. After masking, maximum-intensity projections of identified nuclei were generated in the sfGFP, autofluorescence, and mCherry channels. Cell Profiler was then used to quantify the number, mCherry intensity, GFP intensity, and red (mCherry)/green (sfGFP) ratio of the projected nuclei.

**Arabidopsis AtWUS seedling treatment**. Arabidopsis (Col-0) seedlings were grown in 12-well culture plates with 8–10 seedlings per well in 1 ml of 1x MS media supplemented with 0.5% sucrose and 2.5 mM MES, pH 5.7. Seeds were sterilized by washing in 70% ethanol for 30 s followed by a 15-min incubation in 50% bleach supplemented with 0.5% Tween-20 and rinsed 5x with DI water. Sterilized seeds were stratified in plates at 4 °C for 3 days after plating. Seedlings were grown at 22 °C under 16-h photoperiods for 12 days. To treat seedlings, the liquid media in each well was replaced with control and WUSCHEL treatments. Control wells were refreshed with 1 mL of MS growth media. WUSCHEL proteins were prepared for use by dialysis into 10 mM MES pH 5.7 for 2 h before dilution to their final concentration. Treatment wells were refreshed with 1 mL of 1 or 3 μM of protein (R9-GFP11, GFP11-AtWUS-R9, GFP11-AtWUS) dissolved in MS growth media. After 24 h of treatment, seedlings were frozen in liquid nitrogen and physically disrupted with chrome steel bearing balls.

**RT-qPCR analysis**. Total RNA was extracted from Arabidopsis seedlings using an RNeasy Plant Mini Kit (Qiagen). Extracted RNA quality was confirmed with a NanoDrop UV–Vis Spectrometer. Complementary DNA was synthesized from RNA using an iScript cDNA Synthesis Kit (Bio-Rad). qPCR was run with PowerUP SYBR Green Master Mix (Applied Biosystems), and each reaction was run in triplicate according to the manufacturer's recommendations. Melt-curve analysis was run after qPCR cycling to confirm primer specificity. Relative gene expression was determined using the ddCt method[80] using SAND1 as the reference gene. Relative gene expression was determined from four biological pools each containing 8–10 seedlings. A list of utilized primers is available in Supplemental Data 2. Statistical comparisons of dd$C_T$ values were conducted as done previously[81] using a *t*-test with Holm–Šídák correction in GraphPad Prism 9. The used primers and accession numbers are provided in Supplemental Data 1.

**Sequence alignment**. The sequences for *Arabidopsis thaliana* WUSCHEL (UniProt: Q9SB92), Shoot Meristemless (UniProt: Q38874), *Zea mays* WUS2 (UniProt: A0AAS6), *Homo sapiens* VAX1 (UniProt: Q5SQQ9), OCT4 (UniProt: D5K9R8), and *Drosophila melanogaster* ANTENNAPEDIA (UniProt: P02833) were aligned using UniProt Clustal Omega (https://www.uniprot.org/align). The aligned sequences were then prepared using the ESPript 3.0 web server[82].

**Statistics and reproducibility**. Data obtained from Cell Profiler was processed using a script written in Python 3.9. Using the Cell Profiler data, GFP positive nuclei were counted using a Python script by setting a threshold defined as a one-tailed 99% confidence interval above the mean green/red ratio in the untreated control or water infiltration control of each experiment. Any nuclei with green/red ratio higher than this threshold would be identified as GFP positive. The percentage of sfGFP positive nuclei was calculated by dividing sfGFP-positive nuclei by the total number of mCherry nuclei counted. In experiments where delivery efficiency is used, delivery efficiency is defined by normalizing the percentage of sfGFP-positive nuclei to the 100 μM R9-GFP11 treatment in that experiment. All summary statistics were calculated using Python before export and statistical analysis in GraphPad Prism 9. Kruskal–Wallis non-parametric ANOVA[83] was used for the analysis of multiple comparisons followed by uncorrected Dunn's non-parametric *t*-test unless otherwise noted. Single comparisons were made using a one-sample *t*-test against the normalized value of 1.0. All presented plots were also generated in GraphPad Prism 9. All imaging data were collected with at least five biological replicates spaced across multiple days and batches of plants as experimental repeats. qPCR was experimentally repeated twice and using a new batch of protein for each repeat.

**Reporting summary**. Further information on research design is available in the Nature Portfolio Reporting Summary linked to this article.

## Data availability
Plasmids generated for this study are available through Addgene (#193860-193867 and #202053-202056) upon final release. All source data, including images, related to this study are available from Dryad with the identifier: https://doi.org/10.6078/D1ZB1S. All raw, uncropped, blot, and gel images are available in Supplemental Fig. S14 and also in the Dryad repository. Further correspondence and requests for materials should be addressed to M.P.L.

## Code availability
The code from this study is available in the Zenodo repository with the identifier: https://doi.org/10.5281/zenodo.7272340.

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

## Acknowledgements

We would like to thank the lab of Prof. Savithramma Dinesh-Kumar for provisioning the initial plasmids for sfGFP1-10. We thank the UC Berkeley Macrolab for providing the backbone vector 1B and LIC cloning protocol. We also would like to thank Antonio Del Rio Flores for his advice and support for protein purification. J.W.W. is supported by NSF GRFP. H.J.S. is supported by the DOD through the NDSEG graduate fellowship. N.G. is supported by an FFAR Fellowship. Confocal imaging experiments were conducted at the CRL Molecular Imaging Center, RRID:SCR_017852, supported by the Helen Wills Neuroscience Institute. We would like to thank Holly Aaron and Feather Ives for their microscopy advice and support. We further acknowledge support of a Burroughs Wellcome Fund Career Award at the Scientific Interface (CASI) (to M.P.L.), a Dreyfus foundation award (to M.P.L.), the Philomathia foundation (to M.P.L.), an NIH MIRA award (to M.P.L.), an NIH R03 award (to M.P.L.), an NSF CAREER award (to M.P.L.), an NSF CBET award (to M.P.L.), an NSF CGEM award (to M.P.L.), a CZI imaging award (to M.P.L.), a Sloan Foundation Award (to M.P.L.), a USDA BBT EAGER award (to M.P.L), a Moore Foundation Award (to M.P.L.), an NSF CAREER Award (to M.P.L), and a DOE Office of Science grant with award number DE-SC0020366 (to M.P.L.). M.P.L. is a Chan Zuckerberg Biohub investigator, a Hellen Wills Neuroscience Institute Investigator, and an IGI Investigator.

## Author contributions

J.W.W. conceived and designed the project. J.W.W. carried out the blotting, imaging, developed the image analysis pipeline, peptide sequence design, protein purification, and the majority of the molecular cloning for this project. J.W.W. wrote the manuscript and prepared the figures. H.J.S. aided in protein purification and designed and executed the majority of the qPCR assays. N.G. aided in experimental prioritization and guided data analysis. H.M.N. aided in molecular cloning. E.L. optimized and performed the dot blotting experiments. C.W. executed qPCR. E.G.-G contributed key project input and troubleshooting. M.P.L. provided technical guidance, research prioritization, and significant manuscript writing input. All authors provided editorial guidance on the manuscript.

## Competing interests

The authors declare no competing interests.

## Ethics

Due diligence was taken to ensure all co-authors were properly credited for their contributions to this work.
