## [Peer Review File · Communications Biology]

Reviewers' comments:

Reviewer #1 (Remarks to the Author):

Authors design and test the use and efficiency of protein delivery system to Tobacco leaf mesophyll cells. The system consists of two parts; the partial GFP fragment that was infiltrated into cells through regular agrobacterium-based transfection. These transfected cells were later incubated with a smaller 16aa complementing GFP peptide to produce fluorescence. Based on the earlier work in animals and plants, authors tested 3 efficacy of 3 cell-penetrating peptides (CPPs) to identify the R9 peptide as effective which was further used to deliver larger proteins such as WUSCHEL and mCHERRY. In addition to fluorescence complementation, authors also use molecular analysis of WUS target genes to confirm the functionality of the exogenously delivered WUS. Though the use of CPPs has been demonstrated earlier in animals and plant systems, the delivery of full-length WUS and mCHERRY is interesting and opens up possibilities of introducing gene-editing proteins and also morphogenetic regulators such as WUS and others to improve regeneration. I have a few questions that need to be addressed, and suggestions that may improve the manuscript.

1. It appears that R9 is not necessary to send the full-length mCHERRY into cells (Fig. 5B, E) the relative fluorescence remains the same. Do the authors have an explanation for why larger proteins do not need R9 while the smaller proteins (fig. 3A) require them? It is also puzzling why authors do not pick up the nuclear fluorescence of mCHERRY from the transfected construct just like in many other cases (Fig. 1B, 4B, 7A). Looking at the pics, I think the fluorescence of full-length mCHERRY is weak to mask the nuclear fluorescence from the transfected construct.

2. Related to the previous point, I do not see authors have tested R9 control in WUS transportation assays. In the absence of this data, it is difficult to judge its requirement. Though the authors find WUS-R9 complementation is predominantly found in some round bodies that could be nuclei. It is essential to confirm this with independent nuclear labels such as DAPI or any other nuclear stains. This is because cells in tobacco leaves produce large background fluorescence that resembles nuclei but they could be chloroplasts and they are normally situated in a different focal plane.

3. Fig. 7D. which shows WUS-mediated transcriptional changes done in Arabidopsis leaves whereas all visualization was done in Tobacco. Since authors make a dramatic claim that full-length WUS protein can enter cells, it is essential to test the expression response of well-characterized WUS target genes such as repression of differentiation promoting transcription factors-KANADI 1, KANADI2, YABBY3, ASYMMETRICLEAVES2 (Yadav et al., *Molecular Systems Biology* (2013)9:654).

4. Fig. 4, the authors link the transient DCIP response to the half-life/instability of the R-GFP11 peptide. However, re-treatment with R-GFP11 peptide did not provide the robust fluorescence complementation observed in the first treatment. Therefore, It appears that instability is not the sole cause of fluorescence decay and there must be other reasons. Can the authors provide an explanation/s?

Minor points.

1. Introduction is rather filled liberally with sentences related to plant cell walls which are not quite true. Line 38: cell wall is stated to be a big barrier. I am not sure it is, in fact, the cell wall is extremely porous and permeable to molecules in the range of 30-60KDa. Page 71, only differentiated cells have large central vacuoles but not many other cell types such as meristematic cells.

Reviewer #2 (Remarks to the Author):

The manuscript from Wang and colleagues presents a new method, named DCIP for delivered complementation in planta, that can be used to assess the efficiency of peptide-mediated protein delivery in plants. Strikingly, it also shows elegantly, and for the first time, that peptide-mediated

delivery can be used to deliver active transcription factor, such as the regulator of stem cell identity WUSCHEL, into plant cells.

The manuscript, which is well supported experimentally and well written, represent a clear advancement in the field of protein delivery in plants. It provides a robust method to assess the efficiency of peptide-mediated protein delivery which is likely to be used by the community. It also provides new and very exciting perspectives by showing that a transcription factor like WUSCHEL, which is a key used in regeneration systems (together with babyboom), can be efficiently introduced in plants without transformation. Although it is less important, for the paper, the experiment showing that LifeAct can be introduced in plant cells using this system and be used to tag actin filaments is also very elegant.

I only have few remarks about this work, that I would like that author to address:

1. I find the number of experimental repeats quite low. They usually are only of 3 to 5 plants. Were these plants harvested at the same time or are they true independent experiments (performed at different times)? Or are they just biological replicates? The standards nowadays would be to have at least two independent experiments with several biological replicates (plants) for each experiment.
2. Related to my previous comment, I am not very convinced by the dotblot shown in Fig. 4.C (even-though it is quantified). The signal of plant 1 and 2 seems to disappear at 8h while in plant 3, it disappears at 4h but reappear at 8h. I think that it would be more convincing if another replicate of this experiment could be shown. In addition, the author also make the hypothesis that the signal disappears fast because the R9-GFP11 that is not internalized is degraded in the apoplast. Couldn't they just test this hypothesis by looking at the dynamics of disappearance of a full R9-GFP (or of their R9-mCherry even if it also have GFP11 attached) in the apoplast?
3. The authors do not see convincing delivery with BP100 at 100 μm , which is a bit surprising but did they try higher concentration.
4. As the authors point in their manuscript, there are some limitations in imaging the delivery of the GFP11-mCherry-R9 recombinant protein using the DCIP as the system already has a nuclear mCherry. Why didn't the author used another fluorophore (like BFP). Wouldn't it work better?

1) biological and technical replicates for each experiment,

We have increased the number of biological replicates for all imaging experiments by two. For other experiments at least two fully independent experiments were conducted.

2) appropriate controls,

We have clarified instances where plastid autofluorescence may obscure the imaging results by including a plastid overlay. We have also tested the delivery of WUS without R9.

3) testing the expression response of well-characterized WUS target genes,

We have retested expression response of 6 direct targets of WUS.

4) imaging the delivery of the recombinant protein with another fluorophore.

We demonstrate the delivery of BFP in our revision.

Reviewers' comments:

Reviewer #1 (Remarks to the Author):

Authors design and test the use and efficiency of protein delivery system to Tobacco leaf mesophyll cells. The system consists of two parts; the partial GFP fragment that was infiltrated into cells through regular agrobacterium-based transfection. These transfected cells were later incubated with a smaller 16aa complementing GFP peptide to produce fluorescence. Based on the earlier work in animals and plants, authors tested 3 efficacy of 3 cell-penetrating peptides (CPPs) to identify the R9 peptide as effective which was further used to deliver larger proteins such as WUSCHEL and mCHERRY. In addition to fluorescence complementation, authors also use molecular analysis of WUS target genes to confirm the functionality of the exogenously delivered WUS. Though the use of CPPs has been demonstrated earlier in animals and plant systems, the delivery of full-length WUS and mCHERRY is interesting and opens up possibilities of introducing gene-editing proteins and also morphogenetic regulators such as WUS and others to improve regeneration. I have a few questions that need to be addressed, and suggestions that may improve the manuscript.

1. It appears that R9 is not necessary to send the full-length mCHERRY into cells (Fig. 5B, E) the relative fluorescence remains the same. Do the authors have an explanation for why larger proteins do not need R9 while the smaller proteins (fig. 3A) require them? It is also puzzling why authors do not pick up the nuclear fluorescence of mCHERRY from the transfected construct just like in many other cases (Fig. 1B, 4B, 7A). Looking at the pics, I think the fluorescence of full-length mCHERRY is weak to mask the nuclear fluorescence from the transfected construct.

The 60 μ M of mCherry used in the infiltration is highly fluorescent to the degree that it will even engage the microscope's emergency safety shutoff if imaged at the laser power we typically use. As such, we had to image with significantly less laser power making it difficult to obtain clear definition of nuclei. However, we agree that a clearer view of nuclear-localized mCherry is warranted so we very carefully reviewed the images used in figure 5 and found clear examples

of nuclear-localized mCherry that also had GFP fluorescence and have included an example of such an event in the figure.

The superficial appearance of the no-CPP mCherry being internalized is due to a combination of the diffraction limited resolution of confocal microscopy and the unevenness of a leaf as an imaging surface. In XY-space, extracellular mCherry will emit light that will spatially colocalize with cytosol. In the axial dimension, extracellular mCherry above and below the focal plane will produce signal that seems to be intracellular. For example, mCherry proteins adsorbed to depressed regions of the leaf will appear in the same imaging plane as slightly raised portions of the leaf and thus give an appearance of internalization. This exact difficulty in quantifying or even identifying uptake events was what led to our development of DCIP. To clarify this situation, we have included panels with higher zoom to demonstrate how one might detect uptake in the absence of DCIP.

2. Related to the previous point, I do not see authors have tested R9 control in WUS transportation assays. In the absence of this data, it is difficult to judge its requirement. Though the authors find WUS-R9 complementation is predominantly found in some round bodies that could be nuclei. It is essential to confirm this with independent nuclear labels such as DAPI or any other nuclear stains. This is because cells in tobacco leaves produce large background fluorescence that resembles nuclei but they could be chloroplasts and they are normally situated in a different focal plane.

We have conducted experiments with WUS lacking a CPP and to our surprise found it cell penetrating (in contrast to either mCherry or BFP without a CPP which do not penetrate). However, on review of the literature we found substantial research showing that homeodomain proteins are generally cell penetrating across all eukaryotic kingdoms. We greatly appreciate the revision comment as these new findings have expanded the scope and impact of the work: the discovery that plant transcription factors might contain conserved domains that are inherently cell penetrating.

Finally, we have included a chloroplast overlay to clarify that the green objects observed are indeed nuclei in the representative micrograph.

3. Fig. 7D. which shows WUS-mediated transcriptional changes done in Arabidopsis leaves whereas all visualization was done in Tobacco. Since authors make a dramatic claim that full-length WUS protein can enter cells, it is essential to test the expression response of well-characterized WUS target genes such as repression of differentiation promoting transcription factors-KANADI 1, KANADI2, YABBY3, ASYMMETRICLEAVES2 (Yadav et al., Molecular Systems Biology (2013)9:654).

We repeated the qPCR experiment using only genes that have been demonstrated as direct targets of WUS: ARR6, CLV3, KANADI1/2, ASL2, and YAB3 using RNA previously purified in the original manuscript and found that all genes expected to be downregulated by WUS (ARR6, KAN1/2, ASL2, and YAB3) were downregulated by WUS treatment and that CLV3 was slightly

upregulated as expected. Furthermore, we repeated the experiment again using a separate batch of protein purified at a later date to demonstrate the experiment's reproducibility.

4. Fig. 4, the authors link the transient DCIP response to the half-life/instability of the R-GFP11 peptide. However, re-treatment with R-GFP11 peptide did not provide the robust fluorescence complementation observed in the first treatment. Therefore, It appears that instability is not the sole cause of fluorescence decay and there must be other reasons. Can the authors provide an explanation/s?

The instability of the DCIP signal is likely a combination of several factors including apoplastic clearance of the peptide and the natural half-life of intracellular peptide and GFP protein. Although we do not have a clear explanation for why re-challenge with the peptide results in seemingly less complementation, we hypothesize that we may be eliciting stress responses in leaf tissue that lead to either higher extra/intracellular turnover or perhaps even a decrease in the cell-wall permeability from stress-induced callose deposition. Based on our peptide stability data in figure 4C and prior published results on CPP instability with extracellular proteases (Palm, Jayamanne et al. 2007, Youngblood, Hatlevig et al. 2007), these assumptions remain hypothetical but could be other reasons for the fluorescence decay. Understanding the phenomena may become important in future attempts to leverage protein delivery in plants and indeed ongoing experiments in our lab is studying the effect of stabilizing peptides with D amino acid substitutions to further elucidate the effect of peptide stability on *in planta* function.

We have also attempted delivery of GFP11 by co-infiltrating with R9 peptide and show that a covalent connection between R9 and its cargo is critical for successful delivery. Although not a complete answer, we show that stability of the construct in general *before* uptake is an important component of successful plant delivery.

Minor points.

1. Introduction is rather filled liberally with sentences related to plant cell walls which are not quite true. Line 38: cell wall is stated to be a big barrier. I am not sure it is, in fact, the cell wall is extremely porous and permeable to molecules in the range of 30-60KDa. Page 71, only differentiated cells have large central vacuoles but not many other cell types such as meristematic cells.

We appreciate the suggestion and have amended the introduction to include this information.

Reviewer #2 (Remarks to the Author):

The manuscript from Wang and colleagues presents a new method, named DCIP for delivered complementation in planta, that can be used to assess the efficiency of peptide-mediated protein delivery in plants. Strikingly, it also shows elegantly, and for the first time, that peptide-mediated delivery can be used to deliver active transcription factor, such as the regulator of stem cell identity WUSCHEL, into plant cells.

The manuscript, which is well supported experimentally and well written, represent a clear advancement in the field of protein delivery in plants. It provides a robust method to assess the efficiency of peptide-mediated protein delivery which is likely to be used by the community. It also provides new and very exciting perspectives by showing that a transcription factor like WUSCHEL, which is a key used in regeneration systems (together with babyboom), can be efficiently introduced in plants without transformation. Although it is less important, for the paper, the experiment showing that LifeAct can be introduced in plant cells using this system and be used to tag actin filaments is also very elegant.

I only have few remarks about this work, that I would like that author to address:

1. I find the number of experimental repeats quite low. They usually are only of 3 to 5 plants. Were these plants harvested at the same time or are they true independent experiments (performed at different times)? Or are they just biological replicates? The standards nowadays would be to have at least two independent experiments with several biological replicates (plants) for each experiment.

In an effort to demonstrate the reproducibility of the experiments, we have increased the biological repeat count of all data by an additional two repeats. The data of each experiment now comprise of no-less than five and in some cases as many as 7 biological replicates. For all experiments, data were collected across multiple days. Typically, a single day of imaging would only encompass 1-2 replicates as the imaging is time consuming. These repeats ensure that our data and reported errors comprise both the experimental variability and biological variability of our assays.

2. Related to my previous comment, I am not very convinced by the dotblot shown in Fig. 4.C (even-though it is quantified). The signal of plant 1 and 2 seems to disappear at 8h while in plant 3, it disappears at 4h but reappear at 8h. I think that it would be more convincing if another replicate of this experiment could be shown. In addition, the author also make the hypothesis that the signal disappears fast because the R9-GFP11 that is not internalized is degraded in the apoplast. Couldn't they just test this hypothesis by looking at the dynamics of disappearance of a full R9-GFP (or of their R9-mCherry even if it also have GFP11 attached) in the apoplast?

We explain the reappearance of the signal at 8H in one case possibly because of uneven infiltration or drying of the leaf. The experiment was conducted in such a way that a single leaf was infiltrated at n-hours before simultaneously harvesting each sample. As such, there may leaf-region specific variation in either the recovery of the R9-GFP11 peptide, its clearance, or its infiltration. We have also repeated the dot blot experiment to the same result and included this data.

We believe that understanding the apoplastic clearance rate of treating the leaf with R9-GFP full-length would be difficult as we would be unable to determine the loss of protein as the result of uptake and intracellular degradation or apoplastic degradation. We agree that the data as presented simply show that the stability of delivered peptide in leaf *tissue* in general is low because of this inability to distinguish intracellular from apoplastic degradation, and have adjusted the wording in the manuscript to suit this. We have also added data showing that non-

covalently attached R9 and GFP11 do not achieve delivery, demonstrating that R9 and its cargo must be somehow attached to be effective.

3. The authors do not see convincing delivery with BP100 at 100 μm , which is a bit surprising but did they try higher concentration.

We did not try higher concentration, however at 100 μm , R9GFP11 reaches near saturation-levels of delivery and the concentration used is many fold higher than typical. From a practical perspective, even if higher concentrations of BP100 were to work, the practicality of expressing and purifying enough protein for delivery with $> 100 \mu\text{m}$ protein concentrations makes BP100 impractical for our system which requires the protein cargo to be expressed with the CPP tag. Because the uptake characteristics of CPPs are likely to be cargo dependent, we've elaborated on this aspect of CPPs in the discussion.

4. As the authors point in their manuscript, there are some limitations in imaging the delivery of the GFP11-mCherry-R9 recombinant protein using the DCIP as the system already has a nuclear mCherry. Why didn't the author used another fluorophore (like BFP). Wouldn't it work better?

We had previously attempted purification of BFP-R9 with no success due to its low solubility in low NaCl buffers that are tolerated by plants. With significant optimization we have purified a small quantity of BFP-R9 and showed its delivery and co-localization with nuclei showing GFP complementation and have included that data. However, rates of delivery were low and difficult to quantify using the DCIP method, likely due to the protein's low solubility below pH 10 and the relatively acidic conditions of plant extracellular space.

Reviewers' comments:

Reviewer #1 (Remarks to the Author):

The authors have largely addressed all the comments raised by this reviewer through new experiments and analysis. These efforts have improved the manuscript.

Reviewer #2 (Remarks to the Author):

In this revision of their manuscript, the authors have answered to my comments. They have even provided more data supporting that WUS itself can enter in the cells without the need of a R9 peptide, which was unexpected. They suggest that the ability of WUS to enter into the cell is associated to the presence of an alpha-helix which is conserved in homeodomain transcription factor. However, as they do not test if the presence of this alpha-helix is necessary (by removing it) and sufficient (by fusing it to GFP11) for WUS penetration into cells, I think that they should be more careful about their statement on WUS internalization. In addition, the work from Daum and colleagues (2014, PNAS) shows that WUS cannot move from cells to cells in the meristem when plasmodesmata are closed, supporting the WUS normally moves through plasmodesmata and not through the cell wall. I think that the authors should discuss the discrepancies between their work and this one.

Reviewer 2 Comment:

In this revision of their manuscript, the authors have answered to my comments. They have even provided more data supporting that WUS itself can enter in the cells without the need of a R9 peptide, which was unexpected. They suggest that the ability of WUS to enter into the cell is associated to the presence of an alpha-helix which is conserved in homeodomain transcription factor. However, as they do not test if the presence of this alpha-helix is necessary (by removing it) and sufficient (by fusing it to GFP11) for WUS penetration into cells, I think that they should be more careful about their statement on WUS internalization. In addition, the work from Daum and colleagues (2014, PNAS) shows that WUS cannot move from cells to cells in the meristem when plasmodesmata are closed, supporting the WUS normally moves through plasmodesmata and not through the cell wall. I think that the authors should discuss the discrepancies between their work and this one.

We performed an experiment by purifying a GFP11-AtWUS mutant with the third alpha helix (AA82-102) deleted and treating plants with this protein ($\Delta\alpha3$). We found that $\Delta\alpha3$ possesses compromised cell penetrating ability (51% for WT GFP11-AtWUS and 15.7% for $\Delta\alpha3$). These experiments show that the third helix of the WUS homeodomain is important for the cell penetrating efficacy of the WT protein as it is in other homeoproteins. It is unclear to us why a low, basal amount of delivery was observed in $\Delta\alpha3$ but this could be due to other unidentified cationic sequences or from non-specific uptake from the concentration of protein used in the experiment.

An important distinction between our results and those from Daum and others investigating WUS intercellular trafficking is that we are applying the protein exogenously and in fairly high concentration thus bypassing plasmodesmata barriers and overcoming the cell wall barrier by simple diffusion as the cell wall is permeable to small macromolecules. Therefore, the controlled intercellular plasmodesmata trafficking previously observed may not be relevant to our system as we are mostly interested in whether exogenous delivery of WUS is possible. We have included a statement of this finding in our manuscript.

Finally, although this is far outside the scope of our manuscript and our expertise, as a matter of speculation, the presence of plasmodesmata (PD) in plants would suggest that homeoproteins do not need prior secretion and cell penetration as they do in mammalian cells to act as paracrine signals (Lee, 2019 *Cell Reports*). However, the PD are also tightly regulated membrane lined channels (Knox, 2015, *Plant Physiol.*) and perhaps the cell penetrating characteristics of homeoproteins still plays a role in their intracellular movement through PD via membrane interactions or even desmotubules. The membrane interacting characteristics of homeodomains may explain observations by Daum *et al* that show the WUS homeodomain itself is necessary and sufficient for PD based trafficking. The cell penetrating ability may also be simply evolutionary anachronism or coincidental with the DNA binding motif itself. Our work may signal to experts in this field to explore the potential importance of homeodomains as paracrine signals in plants. With this said, given the lack of direct evidence we feel that inclusion of this speculation is inappropriate in this manuscript but seems worthy of future investigation.